# Structural analysis of phosphoribosyl-transferase-mediated cell wall precursor synthesis in *Mycobacterium tuberculosis*

Shan Gao[1,2,6], Fangyu Wu[1,2,6], Sudagar S. Gurcha[3], Sarah M. Batt [3], Gurdyal S. Besra [3] ✉, Zihe Rao [1,2,4] ✉ & Lu Zhang [2,5] ✉

In *Mycobacterium tuberculosis*, Rv3806c is a membrane-bound phosphoribosyltransferase (PRTase) involved in cell wall precursor production. It catalyses pentosyl phosphate transfer from phosphoribosyl pyrophosphate to decaprenyl phosphate, to generate 5-phospho-β-ribosyl-1-phosphoryldecaprenol. Despite Rv3806c being an attractive drug target, structural and molecular mechanistic insight into this PRTase is lacking. Here we report cryogenic electron microscopy structures for Rv3806c in the donor- and acceptor-bound states. In a lipidic environment, Rv3806c is trimeric, creating a UbiA-like fold. Each protomer forms two helical bundles, which, alongside the bound lipids, are required for PRTase activity in vitro. Mutational and functional analyses reveal that decaprenyl phosphate and phosphoribosyl pyrophosphate bind the intramembrane and extramembrane cavities of Rv3806c, respectively, in a distinct manner to that of UbiA superfamily enzymes. Our data suggest a model for Rv3806c-catalysed phosphoribose transfer through an inverting mechanism. These findings provide a structural basis for cell wall precursor biosynthesis that could have potential for anti-tuberculosis drug development.

5-Phospho-α-ribosyl-1-pyrophosphate (PRPP) is a central metabolite utilized by all domains of life. It acts as an activated form of ribose-5-phosphate involved in cellular biosynthesis of purine and pyrimidine nucleotides, the amino acids histidine and tryptophan, and the cofactors NAD and NADP. All these reactions that utilize PRPP as a donor substrate are catalysed by a group of intracellular enzymes collectively termed phosphoribosyl transferases (PRTases)[1-4]. In the *Mycobacterium* genus, which includes the deadly human pathogen *Mycobacterium tuberculosis* (*Mtb*), PRPP is unexpectedly utilized in the biosynthesis of two key components of the bacterial cell wall, termed arabinogalactan (AG) and lipoarabinomannan (LAM)[5,6]. Specifically, PRPP serves as a pentosyl phosphate donor catalysed by a unique membrane-bound PRTase Rv3806c, which is responsible for bringing the soluble phosphoribosyl moiety to the membrane-anchored decaprenyl phosphate (DP) to generate 5-phospho-β-ribosyl-1-phosphoryldecaprenol (DPPR), a precursor of β-arabinosyl monophosphodecaprenol (DPA)[7-12] (Fig. 1a,b). The latter is the only known arabinosyl donor utilized by a series of glycosyltransferase-C superfamily enzymes in both AG and LAM biosynthesis, including the EmbA/B/C proteins, the known targets of the first-line anti-tuberculosis

[1]State Key Laboratory of Medicinal Chemical Biology, College of Life Sciences, College of Pharmacy, Nankai University, Tianjin, China. [2]Shanghai Institute for Advanced Immunochemical Studies, School of Life Science and Technology, ShanghaiTech University, Shanghai, China. [3]Institute of Microbiology and Infection, School of Biosciences, University of Birmingham, Edgbaston, Birmingham, UK. [4]Laboratory of Structural Biology, Tsinghua University, Beijing, China. [5] Shanghai Clinical Research and Trial Center, Shanghai, China. [6]These authors contributed equally: Shan Gao, Fangyu Wu. ✉e-mail: g.besra@bham.ac.uk; raozh@tsinghua.edu.cn; zhanglu1@shanghaitech.edu.cn

**Fig. 1 | The central role and function of Rv3806c in DPA biosynthesis and overall structure of Rv3806c. a**, PRPP is utilized in both cellular metabolism and cell wall biosynthesis in *Mtb*. **b**, Rv3806c is involved in the cell wall biosynthesis by catalysing phosphoribose transfer from PRPP to a membrane-anchored substrate DP to generate DPPR, an essential precursor of arabinosyl donor DPA for AG and LAM biosynthesis. **c**, [14C] radiolabeled cell-free PRTase activity of purified Rv3806c protein. Data presented are mean ± standard deviation calculated from three independent experiments (Source Data Fig. 1). **d,e**, EM density map (**d**) and ribbon representation (**e**) of Rv3806c trimer reconstituted in nanodisc, coloured by protomer, lipid POPG was coloured in grey. PG, peptidoglycan; MA, mycolic acids.

drug ethambutol (EMB)[13–18]. Hence, blocking cell wall arabinan synthesis by inhibiting the DPA pathway is an ideal approach towards new drug development[19–21]. Given the fact that *rv3806c* is essential for the viability of *Mtb* and the lack of its homologues in humans, Rv3806c is considered as an attractive target in anti-tuberculosis therapeutics[7,9]. In the clinic, *rv3806c* mutations were found to be involved in EMB resistance in *Mtb* without common *emb* mutations[22–25]. This was presumably caused by elevated DPA levels[24,25], although the molecular basis remains unclear.

The molecular basis of water-soluble PRTases has been widely described[3,26–31]. However, Rv3806c is distinct in its intramembrane nature compared with other reported PRTases and does not share obvious sequence similarities with the latter[27]. Despite its central role in *Mtb* cell wall biosynthesis, the structure and molecular mechanism of Rv3806c or any of its mycobacterial homologues remains unknown. Revealing the membrane-bound architecture of Rv3806c is critical in elucidating how it recognizes the solvent soluble PRPP and transfers phosphoribosyl moiety to the membrane soluble DP to generate the key DPR/DPA precursors. In this Article, we report the cryogenic electron microscopy (cryo-EM) structures of Rv3806c from *Mtb* in complex with its donor PRPP and endogenous acceptor DP substrates, respectively. Our study provides an important framework for the mechanistic understanding of the function of this unique membrane-bound PRTase.

## Results

### Functional and structural characterization of Rv3806c

FLAG-tagged Rv3806c was expressed in *M. smegmatis* (*Msm*) and purified to homogeneity. Rv3806c in glyco-diosgenin (GDN) solution was used for initial electron microscopy (EM) analysis and functional studies (Extended Data Fig. 1a–c). Negative-stain EM analysis of Rv3806c in GDN revealed a threefold symmetry assembly (Extended Data Fig. 1i). Using a previously established cell-free PRTase assay[32], we have shown that purified Rv3806c is enzymatically active and the PRTase activity is not inhibited by EMB (Fig. 1c and Extended Data Fig. 1c), indicating that Rv3806c is not a molecular target of EMB. The protein was then reconstituted into nanodisc for structure determination by cryo-EM single-particle analysis (Extended Data Fig. 1d,e). Two-dimensional classification reveals a clear threefold symmetry (Extended Data Fig. 2b), consistent with the sample purified in GDN solution. This allowed us to improve the resolution to an overall 3.36 Å upon imposing a C3 symmetry in the final refinement step. The density is of good quality, allowing accurate modelling of most residues of Rv3806c including 18–78 and 93–302. The sidechains of 79–92 could not be traced and were modelled as poly-Ala, while the 1–17 residues of the N-terminal were not modelled due to flexibility. Also clearly resolved was the major part of the endogenous substrate DP, which will be discussed later in detail (Fig. 3c, Extended Data Fig. 4a and Supplementary Information).

### Rv3806c resembles a UbiA superfamily fold

Each Rv3806c protomer consists of nine transmembrane (TM) helices connected by four periplasmic loops (PLs) and four cytoplasmic loops (CLs). The TM helices, in a counterclockwise C shape arrangement viewed from the cytosolic side, are divided into two helical bundles, bundle 1 formed by TMs 1–5 and bundle 2 by TMs 6–9, leaving a big cleft between TM 1 and TM 9 (Figs. 1e and 2a,b). CL1 and CL4 form short

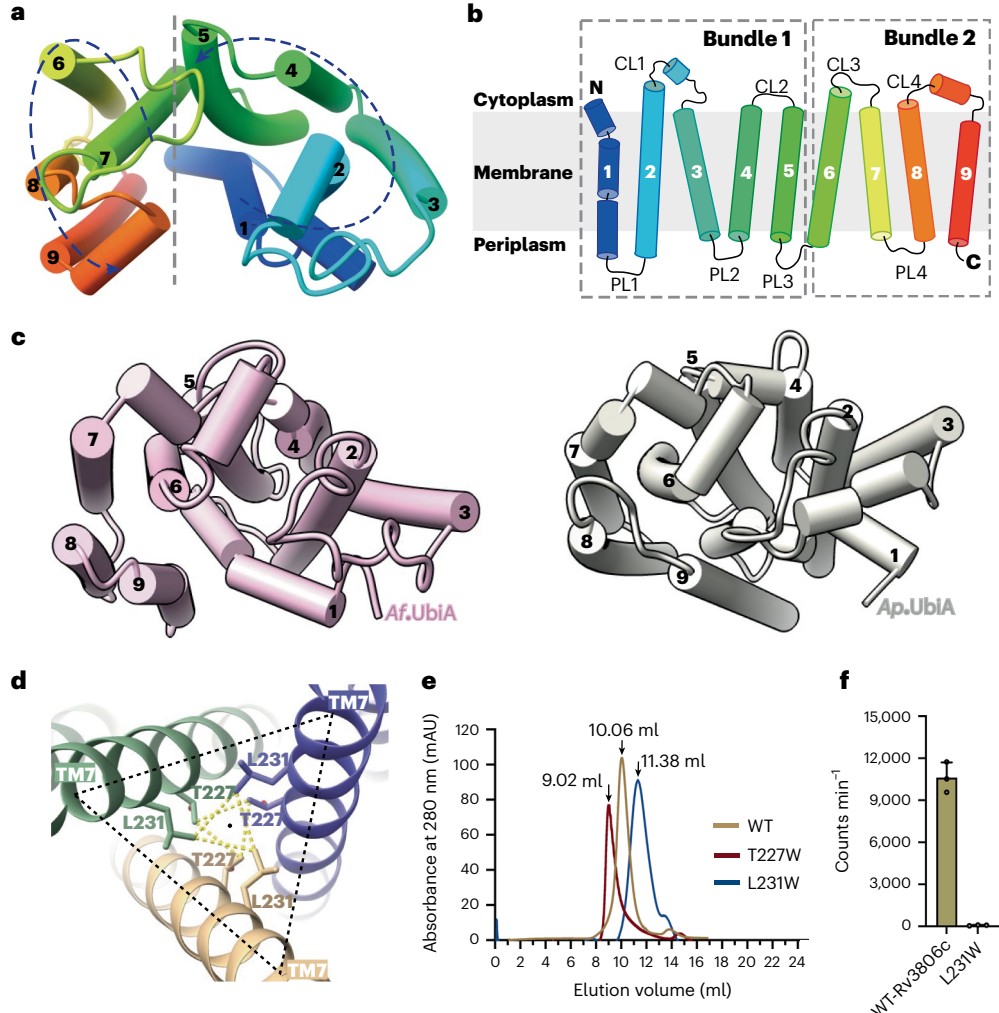

**Fig. 2 | UbiA-like fold of Rv3806c topology and functional importance of trimeric assembly. a,b**, Counterclockwise TM arrangement indicated in dashed line and arrows (**a**), and topology diagram of Rv3806c promoter shows two TM bundles enclosed by two dashed lines (**b**). **c**, TM and cytoplasmic loop arrangemt of *Af*UbiA (left) and *Ap*UbiA (right). **d**, T227 and L231 on TM 7 are central for Rv3806c trimer formation. Residues are shown as sticks, and TM 7 is shown in ribbon representation. **e**, SEC chromatographs of WT-Rv3806c, L231W and T227W mutants, the T227W mutant was eluted close to the void volume. **f**, PRTase activities of WT-Rv3806c and L231W mutant. The data presented for the WT-Rv3806c and L231W mutant lane are mean ± standard deviation calculated from three independent experiments (Source Data Fig. 2).

juxta-membrane helices while the PLs are relatively less ordered. The structure of Rv3806c reveals an overall fold similar to that of the UbiA superfamily, with a $C_\alpha$ root mean square deviation of 7.788 Å across 262 residues to the UbiA from *Archaeoglobus fulgidus* and 8.668 Å across 263 residues to the UbiA from *Aeropyrum pernix*, respectively[33,34]. The similarity refers to nine-TM-helix arrangement and CL1–3 forming a cap above the transmembrane region (Fig. 2c). The UbiA superfamily are intramembrane prenyltransferases that catalyse the prenylation reaction of structurally diverse biomolecules and play a key role in cellular respiration, photosynthesis and cell damage repair[35–39]. This suggests the possibility that the gene encoding the membrane-bound PRTase in *mycobacteria* may have descended from an ancient gene encoding prenyltransferase. The fact that these two enzymes both use a diphosphate-activated donor substrate and a prenyl-containing substrate may cause the evolutionarily preserved similar helical structural arrangements. Despite their folding similarity, they differ mechanistically, which will be discussed later.

## Rv3806c is functional as a trimer

In this study the structure of Rv3806c was resolved as a trimer in nanodisc (Fig. 1d and Extended Data Figs. 2 and 3). The trimer interface is formed in the membrane by TM 7 arranged in a triangle configuration closely surrounding the trimer axis (Figs. 1e and 2d). In particular, residues Thr227 and Leu231 in the middle of TM 7 of one protomer interact with their symmetric residue contacts from the adjacent protomer within a 4 Å distance (Fig. 2d). Close interfaces between two Rv3806c protomers are mediated by intensive interactions on the TM 6/7/8 and CL2 (Extended Data Fig. 5a). Besides direct protein interactions, lipid-like densities were found between the Rv3806c protomer–protomer interface at the periplasmic site (Fig. 1d and Extended Data Fig. 5b). Thus, the Rv3806c forms a compact trimer mediated by residues on the trimer interface, protomer–protomer interface and filled lipids. To test the oligomeric status of Rv3806c in the native membrane environment, we performed structure-guided disulfide cross-bridge experiments as described[40] (Extended Data Fig. 1f,g). The result shows that Rv3806c primarily forms a trimer on the *Msm* membrane, consistent with the EM results (Fig. 1d and Extended Data Figs. 1i and 2b).

To analyse the functional importance of the trimeric assembly, mutations were introduced to Leu231 and Thr227 aimed to destabilize the trimer. When substituted by a bulky sidechain, the L231W mutation was found to shift the equilibrium to favour a lower oligomeric state

as monitored by size-exclusion chromatography (SEC) and blue native polyacrylamide gel electrophoresis (BN PAGE) (Fig. 2e and Extended Data Fig. 1h). Further negative-stain and microfluidic modulation spectroscopy analysis[41] of the L231W mutant in GDN solution revealed a properly folded but smaller size of the mutant sample compared with the WT-Rv3806c sample (Extended Data Fig. 1i,j). Subsequent enzymatic activity assays revealed that the L231W mutation totally abolished PRTase activity (Fig. 2f). While the T227W mutant was eluted close to the void volume of SEC, suggesting an unproperly folded feature as also indicated by negative-stain analysis (Fig. 2e and Extended Data Fig. 1i). These results provide the molecular evidence for the importance of oligomeric state in PRTase function.

## DP binding site

The structure of Rv3806c reveals a cavity in the TM region that is connected to an extramembrane cleft opening to the cytoplasm (Fig. 3a). The cavity, formed by TM 1 of helical bundle 1 and all the TM helices of helical bundle 2, is amphipathic with a lower hydrophobic portion deep inside the membrane and a upper positively charged part. This cavity is accessible to the inner leaflet of membrane via a cleft created by TM 1 and TM 9 and was hypothesized as an ideal entrance for the lipid-soluble substrate DP (Fig. 3b). Interestingly, during enzymatic analysis, a notable activity was repeatedly detected in the control group in the presence of enzyme but absence of exogenous DP (Fig. 1c). This raised a possibility that endogenous DP exists in the purified Rv3806c and acts as a substrate in our enzymatic activity assays. Upon careful inspection of the density around the cavity, a continuous density was found compatible with a polyprenyl-phosphate ligand and was modelled as a truncated DP molecule with the last five prenyl groups disordered due to flexibility (Fig. 3c and Extended Data Fig. 4a). The entire DP molecule within the purified Rv3806c was consequently confirmed by liquid chromatography–mass spectrometry analysis in a dose-dependent manner (Extended Data Fig. 1k). The modelled truncated DP allow us to define the primary recognition site of the lipid acceptor of Rv3806c. The phosphate group forms polar interactions with Lys28 on TM 1, Try157 on TM 5 and Lys191 on TM 6, while the prenyl tail forms hydrophobic interactions with residues on TM 6 and TM 8 (Fig. 3d). The bound DP and the membrane cleft observed in this study suggested a lateral access mechanism of the lipid substrate entrance. Mutations of the three polar interaction sites revealed that the positively charged residues are essential for catalysis (Fig. 3e).

Next, we further analysed the difference of prenyl ligand binding cavity between Rv3806c and the UbiA superfamily enzymes. Structural alignment with polyprenyl-bound UbiA enzymes revealed that DP-bound Rv3806c has the largest lateral cleft throughout the membrane bilayer due to the long distance between TM 1 and TM 9 (Fig. 3b,c). The UbiA superfamily enzymes, on the contrary, display an upper PP-prenyl binding cavity with a half inside the membrane bilayer and the other half above the membrane boundary[33,34,42] (Fig. 3c), and the distance between TM 1 and TM 9 in UbiA enzymes is shorter than Rv3806c (Fig. 3b). This is mechanistically reasonable as the binding of PP-prenyl that serves as the donor substrate requires both an intramembrane environment where the product is released and an extramembrane environment where the PP$_i$ group is released. In the geranyl-thioloPP-bound UbiA structure, the GSPP cavity is semi-open laterally to the membrane caused by the bending of TM 1 in the middle away from TM 9 (ref. 43). These structures suggested that the lateral opening of the prenyl binding cavity which is regulated by the first and the last TM helices of the two classes of enzymes is determined by both ligand position and prenyl length.

## Structure of PRPP-bound Rv3806c

Rv3806c functions as a membrane-bound PRTase in a puzzling mechanism different from either soluble PRTases or intramembrane UbiA superfamily enzymes. Mg$^{2+}$ has been reported as a key ligand

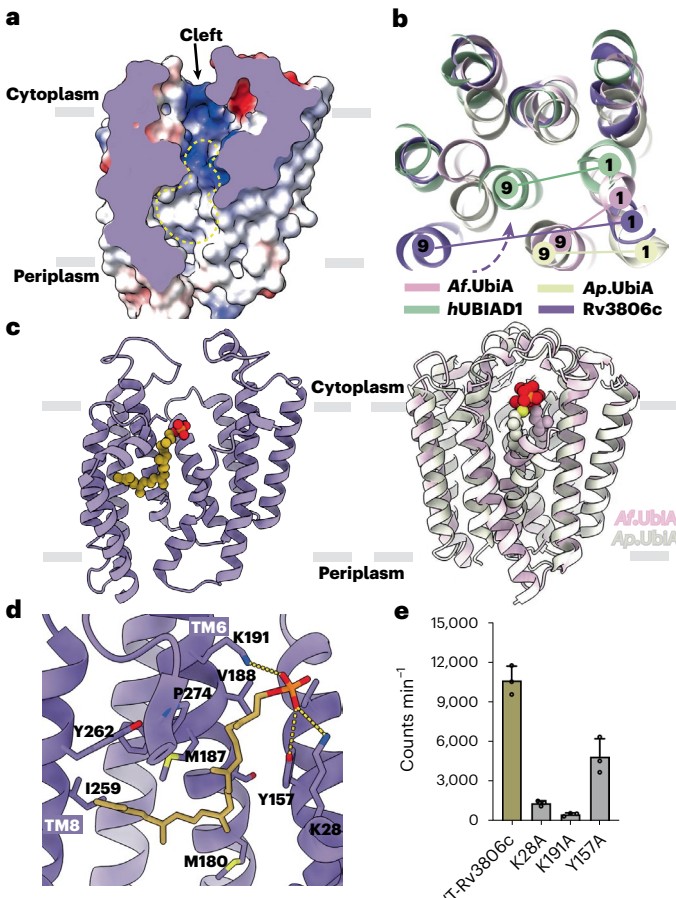

**Fig. 3 | DP binding site. a**, The TM cavity (yellow dashed line) and extramembrane cleft (black arrow) of Rv3806c represented as electrostatic potential surface. **b**, Rv3806c (purple) TM region superimposed onto *Ap*UbiA (yellow), *Af*UbiA (dark pink) and human UbiAD1 (green), the intramembrane cleft formed by TM1 and TM 9 were indicated. The dashed arrow indicates the proposed DP entrance via lateral access from the lipid bilayer. **c**, The structure of DP-bound Rv3806c (left), and structures of PP-prenyl bound *Af*UbiA and *Ap*UbiA (right), proteins are shown in ribbon representation and coloured in purple, pink and white, respectively. The ligands are shown as spheres and coloured by atom: oxygen in red, phosphate in orange and carbon of DP in gold. **d**, DP forms polar interactions with K28, Y157 and K191 by its phosphate group, and non-polar interactions with TM 6 and TM 8 by its prenyl tail. Polar interactions are indicated by dashed lines. **e**, PRTase analysis of polar interaction sites of DP shown in **d**. Data presented for all mutants are mean ± standard deviation calculated from three independent experiments (Source Data Fig. 3).

that mediates PRPP binding to the type I PRTases[3,26,44]. They either utilize a single Mg$^{2+}$ ion at the catalytic site, such as adenine PRTases and orotate PRTases, or two Mg$^{2+}$ ions such as hypoxanthine/guanine PRTases[3,45–49]. Binding affinity of PRPP to purified Rv3806c in the presence of Mg$^{2+}$ was measured at 12.87 μM by microscale thermophoresis (MST; Fig. 4a and Extended Data Fig. 6a). Further biophysical studies revealed that Mg$^{2+}$ plays an essential role in PRPP binding (Extended Data Fig. 6). To gain molecular insights into PRPP recognition in the membrane by Rv3806c, we determined a second structure of Rv3806c in the presence of PRPP-Mg$^{2+}$ at 2.76 Å resolution. The residues from 79 to 92 are more ordered upon PRPP binding, enabling the sidechain modelling in the PRPP-bound structure (Supplementary Information). An excellent density of PRPP was resolved within the extramembrane cleft in each Rv3806c protomer (Extended Data Fig. 4b). The PRPP binding sites are formed by the helical bundle 1 (Fig. 4b). The pyrophosphate group is bound above the membrane boundary defined by the

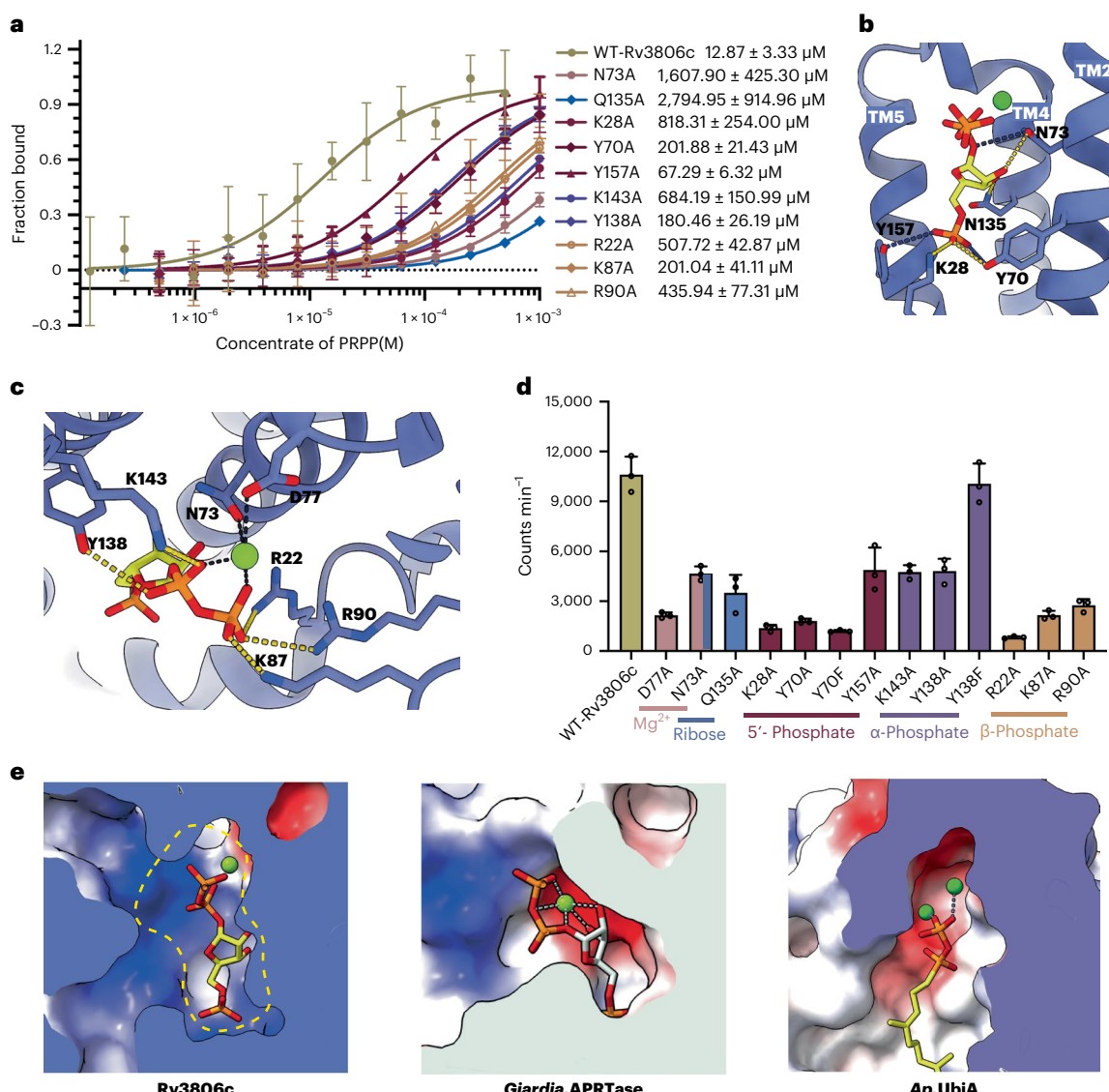

**Fig. 4 | PRPP binding site. a**, Binding affinities of PRPP to WT-Rv3806c and mutant as measured by MST. Data are representative mean values ± standard deviations, calculated from five independent experiments. **b**, Structure of PRPP-bound Rv3806c. Protein is shown in ribbon representation, sidechains of phosphoribose binding sites and PRPP are shown as sticks, and $Mg^{2+}$ is shown as spheres. Yellow dashed lines indicate hydrogen bond with the phosphoribose, and dark-blue dashed lines indicate Van der Waals' force with the phosphoribose based on their distances. **c**, Binding sites of $Mg^{2+}$ and the pyrophosphate group. Black lines indicate the $Mg^{2+}$ coordination involving sidechains and oxygen atoms of the pyrophosphate group. Yellow dashed lines indicate hydrogen bond network with the pyrophosphate group. Sidechains and PRPP are shown as sticks. $Mg^{2+}$ is shown as spheres. **d**, PRTase activity of WT-Rv3806c and mutant, the binding sites of $Mg^{2+}$, ribose, 5'-phosphate and the pyrophosphate were indicated at the bottom. Data presented for all mutants are mean ± standard deviation calculated from three independent experiments (Source Data Fig. 4). **e**, Left: PRPP-$Mg^{2+}$ binding cavities of Rv3806c. Cytoplasmic cap close upon PRPP binding creates the active site indicated as an enclosed dashed line. Middle: PRPP-$Mg^{2+}$ binding cavity of *Giardia* APRTase. Right: GSPP-$Mg^{2+}$ binding cavity of *Ap*UbiA. Cavities are shown as electrostatic potential surface, PRPP and GSPP are shown as sticks, and $Mg^{2+}$ is shown as spheres.

nanodisc belt, similar to the pyrophosphate group bound in the UbiA superfamily. As the phosphoribose group is almost buried inside the TM region, the natural membrane environment prevents the solvent access during the transfer reaction. One molecule of $Mg^{2+}$ was resolved above the pyrophosphate group of PRPP. $Mg^{2+}$ bridges the O3B and O1A atoms of the pyrophosphate group to Asp77 on CL1 and Asn73 on TM 2, respectively, a role similar to the $Mg^{2+}$ commonly observed in adenine PRTases and orotate PRTases (Fig. 4c)[3,45–47,49,50]. These two residues are highly conserved across *mycobacteria* spp. and are also conserved in the UbiA superfamily (Extended Data Figs. 7 and 8). Single point mutations of Asp77 and Asn73 had a 82% and 60% decrease of PRTase activity, suggesting a more profound role of the acidic residue of CL1 in Rv3806c function (Fig. 4d). Unlike those hypoxanthine/guanine PRTases, the

second $Mg^{2+}$ (that is, Mg1 in GHPRT) that acts by coordinating the PRPP ribose hydroxyls and ribose-pyrophosphate group bridging oxygen was not found in Rv3806c near the bound PRPP. Instead, this role is achieved by Asn73 that contacts the ribose-pyrophosphate group oxygen and ribose O2, and Gln135 that contacts the ribose O3 (Fig. 4b). Substitution of Gln135 to Ala leads to a 70% decrease in Rv3806c activity, suggesting the position of ribose is also important in the phosphoribosyl transfer reaction (Fig. 4d). The 5'-phosphate group inserts deeply in the TM region and forms interactions with Lys28, Tyr70 and Tyr157 (Fig. 4b). The first two residues are conserved and important for Rv3806c activity, while Try157 had moderate role in activity and is replaced by Phe in some *mycobacteria* species (Fig. 4d and Extended Data Fig. 7). To be noted, Lys28 has a dual role as both DP and PRPP

binding sites as observed in the DP-bound and PRPP-bound structures, whereas its role in the catalytic transition state remains to be defined. Due to the transfer-group difference between the PRTase and PrenylTase[33,48], the phosphoribose-binding sites are non-conserved between the Rv3806c and UbiA superfamily (Extended Data Fig. 8). The pyrophosphate group is stabilized by a positively charged residue cluster (K/R cluster) including Arg22 on TM1, Lys87 and Arg90 on CL1, and Lys143 on CL2. Tyr138 on TM 5 also participates in pyrophosphate group interaction (Fig. 4c). All these pyrophosphate group binding sites are highly conserved among mycobacteria and play different roles in Rv3806c activity. Specifically, decreased enzymatic activities of mutations of the β-phosphate binding sites (R22A, K87A and R90A lost activity by 93%, 82% and 77%, respectively) had more profound effect than the α-phosphate binding sites (K143A and Y138A lost activity by 60% and 59%, respectively) (Fig. 4d). This suggested that the distal phosphate group of the reaction centre at C1 ribose was more critical in catalysis. Lys143 and Lys208 on helical bundle 2 result in a positively charged part of the extramembrane cleft that enables a direct interaction with the pyrophosphate group, similar to the one-$Mg^{2+}$-involved adenine PRTases and orotate PRTases. The negatively charged pyrophosphate group binding pocket of UbiA enzymes, on the contrary, require the involvement of a second $Mg^{2+}$ for pyrophosphate group coordination (Fig. 4e). Binding affinity analysis revealed that the ribose binding sites override the other groups in PRPP binding, and indicated that the K/R cluster residues and Lys28 play important roles in pyrophosphate group and 5′-phosphate group binding, respectively (Fig. 4a).

The lipid-shaped density is much clearer in the PRPP-bound map and was modelled as palmitoyl-oleoyl-phosphatidylglycerol (POPG) molecule based on the density shape and lipids used for nanodisc reconstitution (Extended Data Fig. 4b). The polar part of POPG heads to the periplasmic surface, while the hydrophobic tails insert into the cleft formed by TMs 4–6 of one protomer and TMs 7–8 of the other. The phosphate group interacts with the mainchain $NH_2$ of Leu172, and the hydroxyl group of the glycerol head forms a hydrogen bond to the sidechain of Ser246 (Extended Data Fig. 5b).

### PRPP binding induces conformational changes

The TM helices remain largely unchanged upon PRPP binding, an exception being the N-terminal half of TM 1, which became a more ordered helix by interaction with the bound PRPP. The CLs undergo conformational changes to different extent in the presence of PRPP. CL1 has a notable conformational change by a horizontal movement towards the bound PRPP-$Mg^{2+}$, while CL3 moves downwards to the ligand direction. CL2 forms an ordered short helix but no visible movement, and CL4 has relatively minor changes upon PRPP binding (Fig. 5a). These ultimately result in a closed extramembrane cleft to block both efflux of the bound PRPP and the influx of solvent into the active site of Rv3806c. This conformational change observed in Rv3806c is similar to the reported closed active site structures of the PRTases mediated by flexible loops only captured in the PRPP binding state, which has been indicated as an essential step for catalysis[3,26,51]. Similarly, closure of the extramembrane cleft upon PRPP binding defines the catalytically active site of Rv3806c (Figs. 4e and 5b). This similarity suggested a common strategy employed by both soluble and membrane-bound PRTases regardless of the difference in their primary structures. The DP molecule was not found in the PRPP-bound structure. This can be explained by the consumption of the endogenously bound acceptor substrate in the reaction when supplied with PRPP-$Mg^{2+}$ at an excess concentration over the enzyme–DP complex.

### Catalytic mechanism of Rv3806c

In this study we found endogenously bound DP within the purified Rv3806c, which indicates a strong binding affinity of the lipidic acceptor substrate to the membrane-bound Rv3806c. In this sense, DP binding stabilizes the enzyme and allows a rapid initiation of the phosphoribose transfer reaction once the donor substrate PRPP is available with access into Rv3806c through the extramembrane cleft from the cytosol. The conformational changes of CLs and TM1 stabilize the bound PRPP in a $Mg^{2+}$-dependent manner and close the active site as a prerequisite of catalysis. Alternatively, the donor-bound structure indicates that the enzyme can exist as a PRPP-bound form without DP; hence, the possibility that PRPP binds before DP cannot be ruled out. The two substrate-bound structures support the inverting mechanism of Rv3806c-mediated reaction: upon superimposing the two structures, the O1 atom of the DP phosphate is in a close distance (4.5 Å) to the C1 anomeric atom of PRPP (Fig. 5a). Most importantly, the DP phosphate is at the opposite side of the donor's leaving group $PP_i$. In such a geometry, the nucleophile attack to the C1 ribose could only be achieved in a back-side attack fashion. This is consistent with the proposed inverting mechanism reaction catalysed by Rv3806c, which turns the donor PRPP ribose in its α configuration to the product DPPR in its β configuration. The membrane-accessible clefts and the extramembrane clefts may enable the release of the products DPPR and PPi, respectively (Fig. 5c). In previously reported purine PRTase studies, an invariant Tyr residue is proposed essential in catalysis, which acts by stabilizing the transition state by forming a cation–π interaction with the ribooxacarbenium ion under the proposed $S_N1$ mechanism[29,51,52]. In the PRPP-bound Rv3806c structure, two Tyr residues, Tyr70 and Tyr138, were found close to the reaction centre. The two Tyr residues are conserved among *mycobacteria* and have similarly moderate contribution in PRPP binding (Fig. 4a and Extended Data Fig. 7), whereas their roles in catalysis differ. When substituted to Ala, Tyr70 shows a more decreased activity (85% loss) than Tyr138 (59% loss) but neither totally abolished activity. When replaced to the aromatic residue Phe, only the activity of Y138F was rescued to 85% of the WT-Rv3806c (Fig. 4d). Moreover, Tyr138 is conserved in the UbiA superfamily, and the equivalent residues in *Ap*UbiA (Tyr115) and *Af*UbiA (Tyr 139) were found essential for catalysis[33,34] (Extended Data Fig. 8). This suggested that Tyr138 may participate in, but be inadequate for stabilizing the transition-state Rv3806c. Some other uncharacterized mechanism may be involved in catalysis. To illustrate the precise substitution reaction mechanism, structural and biochemical studies of the transition state Rv3806c are required.

### Rv3806c mutations associate with EMB resistance

EMB is a front-line antitubercular drug used in the clinic. It has been widely studied that mutations in *embCAB* operon, which encodes the EMB target arabinosyltransferases EmbA/B/C, confer resistance to EMB[53]. Recently, mutations within *rv3806c* were reported to be specifically associated with high-level EMB resistance and some isolates even without common *embB* mutations[13,23–25]. The mutation sites include amino acids on TM 6/7/8 of helical bundle 2, with the most mutations cluster in TM 6. To be noted, all these mutation sites locate at the periplasmic surface of Rv3806c that is far from the active site (Extended Data Fig. 9b), suggesting a possible allosteric regulation mechanism employed in *rv3806c* mutation under EMB stress. Previous studies have shown that EMB competes with DPA to the active site of EmbB[13], which implies that the resistance-associated *rv3806c* mutations may be related to DPA levels. To test this possibility, we introduced clinical resistant mutations of Rv3806c to study the turnover rate of DPPR. Enzymatic assays showed that A249G confers higher DPPR turnover rate than WT-Rv3806c (Extended Data Fig. 9a). Structural analysis revealed Ala249 participates in the interaction with bound POPG, which acts as the lipid glue of the Rv3806c protomer–protomer interface to enhance the trimer stabilization. Specifically, Ala249 sits at the N-terminal end of TM 8, interacts with the hydrophobic tail of POPG and is spatially close to the Ser246, the latter is hydrogen bonded with the glycerol hydrogen of POPG. Hence, we proposed that, with a Gly substitution on Ala249, Rv3806c TM 8 may enhance the interaction with POPG by a possible helical bend towards the bound lipid[54] (Extended Data Fig. 9b).

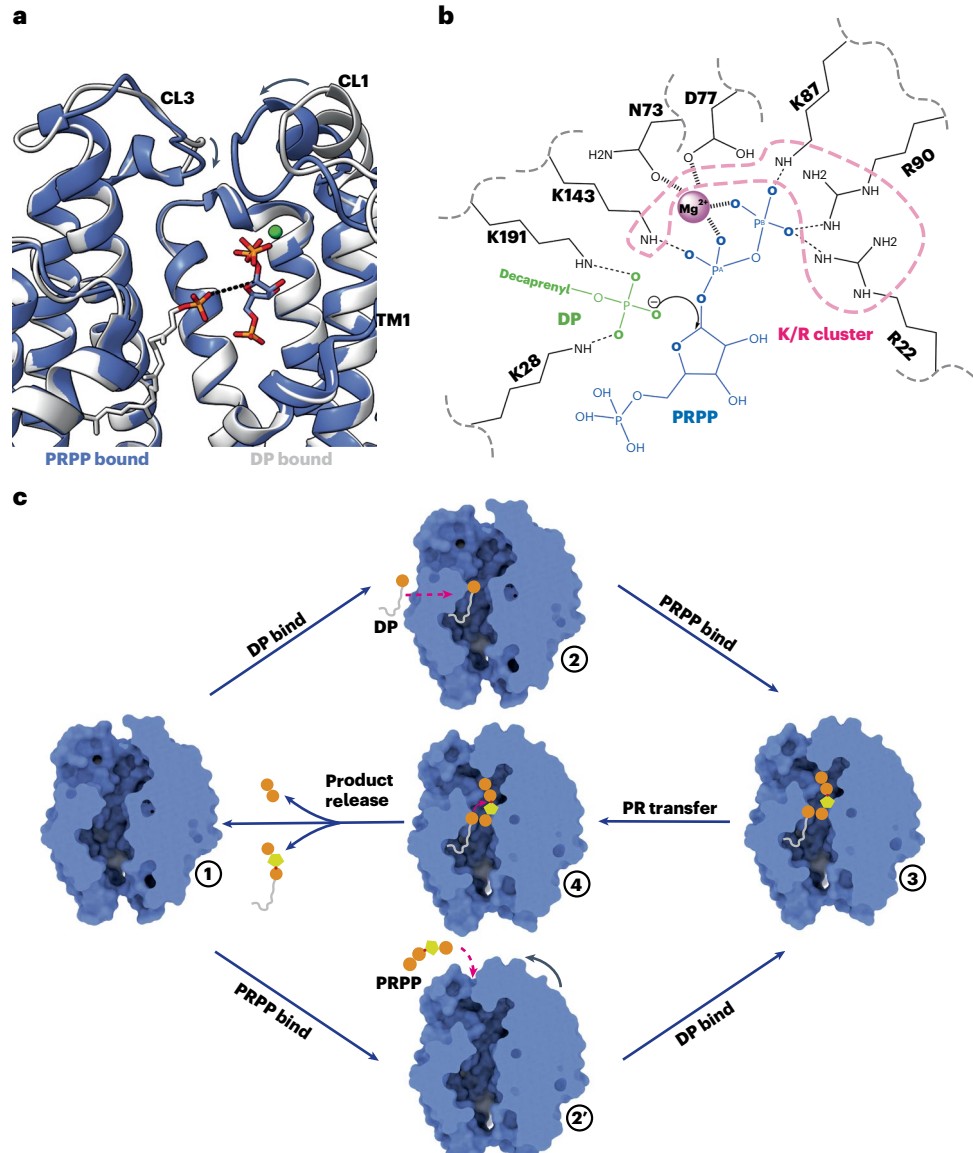

**Fig. 5 | Conformational change of the active site upon substrate binding and proposed catalytic model of Rv3806c. a,** Conformational change of CL1, CL3 and TM1 upon PRPP binding by superimposing the PRPP-bound structure (blue) onto the DP-bound structure (white). The dashed line links the O1 atom of DP phosphate to the C1 anomeric atom of PRPP ribose. PRPP and DP are shown as sticks. The DP phosphate group is close to the C1 ribose and facing opposite to the pyrophosphate group of PRPP. **b,** Schematic representation of the active site. Backbones of interaction sites of Rv3806c are indicated as grey dashed lines, PRPP and DP are coloured in blue and green, K/R cluster including R22, K87, R90 and K143 is circled in pink dashed line. **c,** Proposed catalytic model of Rv3806c. See text for description.

## Discussion

In this study, Rv3806c assembles as an enzymatically functional trimer in the lipid environment. This assembly is proposed to favour a relatively rigid core region formed by helical bundle 2 where the hydrophobic substrate DP is embedded, and a relatively flexible peripheral region formed by helical bundle 1 which undergoes conformational changes upon PRPP binding as observed in this study. The acceptor substrate DP binds in the helical bundle 2 cavity featured by an intramembrane cleft, suggesting a possible lateral access mechanism for lipid substrate entrance regulated by the last and the first TMs. Donor substrate PRPP binding mediated by $Mg^{2+}$ encloses the extramembrane cleft at the cytoplasmic side upon conformational changes of the CLs. The 5′-phosphate and the β-phosphate of the pyrophosphate group binding sites play significant roles in Rv3806c activity, while the $Mg^{2+}$ and ribose are important in PRPP binding. The overall structure of Rv3806c resembles a UbiA superfamily fold in terms of the TM and

cap arrangement, while the DP and phosphoribose binding sites of Rv3806c are different with the reported UbiA family. These structural differences, together with the central role of Rv3806c for *Mtb* cell wall synthesis, make this specific PRTase an attractive target and provide a structural basis for designing novel inhibitors of this process in *Mtb* cell wall biosynthesis.

POPG binding sites on the periplasmic side of Rv3806c map to the clinical EMB resistance mutations, suggesting the importance of this site in Rv3806c function. Given that Rv3806c is the only integral membrane protein in DPA biosynthesis before its periplasm translocation, an alternative possibility was proposed under a 'DPA feedback inhibition' mechanism. In such a proposed scenario, the observed POPG binding sites could serve as an allosteric site for the binding of the end-product DPA, which at its high concentration may compete with the observed phospholipids to achieve feedback inhibition. Further functional studies are required to test this possibility. During

structural analysis, we also found that the inner core of Rv3806c forms a continuous channel almost penetrating through the membrane lipid bilayer. This is a unique feature of Rv3806c not found in other reported structures of the UbiA superfamily enzymes, and raises two possibilities: (1) Rv3806c can bind DP at both sides of the membrane lipid bilayer through this channel and thus effectively function as a DP flippase. It was further speculated that Rv3806c is a dual functional enzyme that effectively couples DP recycling (that is, as a by-product of the reaction catalysed by the EmbA/Aft enzymes) as a flippase and phosphoribosyl transfer as a PRTase. (2) The periplasmic site of the channel may serve as the allosteric site for the possible DP-pentose (DPA or PDR) to modulate the enzymatic function of Rv3806c in the membrane. These possibilities are only tentatively assigned through our structurally studies, but would require further studies to identify the functional mechanism.

## Methods

No statistical method was used to pre-determine sample size. No data were excluded from the analyses. The experiments were not randomized. The investigators were not blinded to allocation during experiments and outcome assessment. The sodium dodecyl sulfate (SDS)–PAGE for Rv3806c purified in detergent and reconstituted in nanodisc was repeated in three independent experiments. The thin-layer chromatography (TLC) autoradiogram for the RPTase activity was performed once. The western blot and blue native PAGE for wild-type (WT) and mutant Rv3806c proteins were repeated in three independent experiments. The negative staining micrograph for WT and mutant Rv3806c samples were repeated in three independent experiments with similar results shown in Extended Data Fig. 1i.

### Protein expression and membrane preparation

The *rv3806c* gene was cloned into the pMV261 vector fused with a flag tag at the N-terminus, under the control of an acetamide promoter. For primers for the molecular cloning of WT and mutant Rv3806c, see Supplementary Data. For the proteins used in MST assay, an extra His-tag was introduced at the N-terminus of the Flag-tag. Recombinant plasmid was introduced into *Msm mc²155* competent cells by electroporation. For large-scale production, cells were cultured in 1 litre Luria broth medium supplemented with 50 μg ml⁻¹ kanamycin, 20 μg ml⁻¹ carbenicillin and 0.1% (v/v) Tween80 at 37 °C with shaking at 220 rpm until the $OD_{600}$ reached 1.0. Four days after induction with 0.2% (w/v) acetamide at 16 °C, the cells were collected in Buffer A containing 20 mM HEPES and 150 mM NaCl, pH 7.5. Cells were lysed by high-pressure homogenizer at 1,200 bar at 4 °C. Cell debris was cleared by centrifugation at 12,000$g$ for 12 min at 4 °C. The membrane pellet was collected by ultracentrifugation at 150,000$g$ for 1.5 h at 4 °C, then resuspended in Buffer A and stored at −80 °C until use. All mutants were overexpressed using the same protocol as the WT protein.

### Protein purification and nanodisc reconstitution

The membrane fraction resuspended in buffer A was incubated with 1% (w/v) *N*-dodecyl-β-ᴅ-maltoside (Anatrace) for 1.5 h at 4 °C. The supernatant separated by centrifugation (40,000$g$, 30 min, 4 °C) was incubated for 1 h with 1 ml anti-Flag resin (GenScript). The beads were then washed with 15 ml Buffer B containing 20 mM HEPES, 150 mM NaCl, 0.02% *N*-dodecyl-β-ᴅ-maltoside, pH 7.5 and then 20 ml Buffer C containing 20 mM HEPES, 150 mM NaCl, 0.04% GDN (Anatrace), pH 7.5 by gravity flow. The protein was eluted in Elution Buffer (20 μg ml⁻¹ 1× Flag peptide in Buffer C), concentrated and further purified by SEC (Superdex 200, 10/300 GL column, GE Healthcare) in Buffer C. The purified protein was then incorporated into lipid nanodisc with a molar ratio 1:3:150 between Rv3806c:MSP1E3D1:POPG (Avanti) and incubated for 1 h with gentle agitation at 4 °C (refs. 55,56). Nanodisc Rv3806c samples were reconstituted by adding bio-beads overnight and further subjected to SEC (Superdex 200) in Buffer A. The peak

fractions corresponding to reconstituted nanodisc samples were collected and concentrated to 2 mg ml⁻¹ for cryo-EM sample preparation.

### Grid preparation and data collection

Three microlitres of nanodisc reconstituted sample was applied to glow-discharged Quantifoil holey carbon grids (Au 300 mesh R1.2/1.3, Solarus Gatan Plasma System $H_2/O_2$ for 40 s). Grids were blotted 3 s and flash-frozen in liquid ethane and cooled in liquid nitrogen using an FEI Mark IV Vitrobot (humidity 100%, temperature 281 K). Images were recorded on a Thermo Scientific Falcon4 direct electron detector at 300 kV at a magnification of 130,000. The post-column Thermo Scientific Selectris X energy filter was used during the collection. Images were recorded in counting mode and binned to a pixel size of 0.96 Å. Automated single-particle data acquisition was performed with E Pluribus Unum data collection software[57], and images were saved as eer format. Defocus values varied from −1.2 to −2.0 μm. Each stack was exposed with a total dose of 60 e⁻ Å⁻² and dose rate is 7.95 e⁻ per pixel per second. For PRPP-bound Rv3806c, 0.8 mM PRPP and 8 mM $MgCl_2$ were added into nanodisc reconstituted samples for 30 min on ice before cryo-EM sample preparation. Grid preparation and data collection were the same as for the DP-bound samples (see also Supplementary Information).

### EM image processing

All dose-fractioned images were motion-corrected and dose-weighted by MotionCorr2 software[58], and their contrast transfer functions were estimated by cryoSPARC[59]. The subsequent steps were all performed in cryoSPARC. For DP-bound state dataset, 3,228,225 particles were picked automatically from 8,894 images. Two-dimensional classification yielded at least 12 representative classes containing 1,526,333 particles. These particles were subjected to ab-initio reconstruction resulting four classes. Particles from the largest two classes were selected for two rounds of heterogeneous refinement. The largest class from heterogeneous refinement was used for non-uniform (NU) refinement, yielding a 4.26 Å initial map. Using this map as a 3D volume template, 1,526,333 particles selected from 2D classification were reused for another three rounds of heterogeneous refinement followed by 3D NU refinement (applying C3 symmetry), resulting in a 3.36 Å density map from 272,777 particles.

For PRPP-bound state dataset, 3,597,451 particles were picked automatically from 9,908 images. Two-dimensional classification yielded at least 12 representative classes containing 1,535,800 particles. These particles were subjected to ab-initio reconstruction resulting five classes. All classes were subjected to heterogeneous refinement. Using the 3.36 Å DP-bound map as a 3D volume template, particles from the largest two classes were selected for another two rounds of heterogeneous refinement followed by 3D NU refinement (applying C3 symmetry), resulting in a 2.76 Å density map from 410,532 particles (see also Supplementary Information).

### Model building and refinement

For DP-bound state dataset, the initial model was built de novo on the basis of an AlphaFold2 predicted structure and then manually adjusted using COOT 0.9.8 (ref. 60). Residues 1–17 were not built. Residues 79–92 were mutated to Ala. The model was subjected to three rounds of real-space refinement using PHENIX 1.17 (ref. 61) and manually adjusted in COOT.

Structure for PRPP-bound Rv3806c complex was built and refined in the same way as the DP-bound structure. Residues 1–12 were not built. Structure figures were generated using PyMOL[62] ChimeraX[63] (see also Supplementary Information).

### Phosphoribosyltransferase activity assays

P[¹⁴C]RPP was prepared enzymatically from [¹⁴C]glucose as described[32]. Each assay was performed in a final volume of 80 μl containing

50 mM MOPS buffer (pH 7.9), P[$^{14}$C]RPP (50,000 cpm), 80 μM DP, 2 μM Rv3806c and 0.08% CHAPS. In some assays, divalent cations (2 mM) and EMB (100 μM) were added. Samples were incubated at 37 °C for 1 h, quenched by the addition of 6 ml of chloroform/methanol/water (10:10:3, v/v/v), and thoroughly mixed. Then 2.65 ml of chloroform and 1.125 ml of water were added to each sample. The resulting bi-phasic mixture was vortexed and centrifuged, and the lower organic layer was recovered, washed thrice using 3 ml of chloroform/methanol/water (3:47:48, v/v/v) and dried. The PRTase control assay (minus WT-Rv3806c + decaprenol phosphate) and the PRTase assay (WT-Rv3806c + decaprenol phosphate) were dried (Extended Data Fig. 1c and Source Data Fig. 1, PRTase assays from experiment 1) and the reaction products resuspended in 100 μl of chloroform/methanol/water (10:10:3, v/v/v) and an aliquot removed for scintillation counting to determine the incorporation of P[$^{14}$C]RPP into the radiolabelled reaction product DPP[$^{14}$C]R. A further aliquot from these two samples was analysed using aluminium backed silica gel 60 F$_{254}$ TLC plates developed in chloroform/methanol/water/ammonium hydroxide (65:25:3.6:0.5, v/v/v/v) and visualized by autoradiography using Kodak BioMAx MR films (Extended Data Fig. 1c and Source Data Fig. 1, PRTase assays from experiment 1). As the control lane (minus WT-Rv3806c + decaprenol phosphate) produced no detectable product, whereas the WT-Rv3806c (+ decaprenol phosphate) sample produced only one product, DPP[$^{14}$C]R by TLC autoradiography (as shown in Extended Data Fig. 1c), the entire assay product for all subsequent PRTase assays for each of the Rv3806c mutants (Source Data Figs. 2–4 and Source Data Extended Data Fig. 9, experiments 1–3) and additional WT-Rv3806c protein samples (Source Data Figs. 1–4, experiments 1–3; and Source Data Extended Data Fig. 9, experiments 1 and 2) were subjected directly to scintillation counting without the need for TLC autoradiography. As per Source Data Figs. 1–4 and Source Data Extended Data Fig. 9, each assay was repeated in triplicate, expect for Extended Data Fig. 9a, where the WT-Rv3806c PRTase activity was performed in duplicate.

## Mass spectrometry

For DP identification, 50 μl of 1 mg ml$^{-1}$ Rv3806c protein in GDN solution was incubated with 350 μl chloroform/methanol (1:1, v/v) then left overnight on ice. The suspension was converted to a bilayer by adding 250 μl chloroform/water (7:3, v/v) the next day. The lower organic phase was pooled after centrifugation and then dried in a speed vacuum concentrator. The dried lipids were re-dissolved in 20 μl chloroform/methanol. Two microlitres of the sample was injected into a quadrupole time-of-flight (SCIEX 4600) mass spectrometer coupled with UPLC (Shimadzu, 30A). After loading the sample onto the chromatography column (ACE Bioresolve Polyphenyl, 300 Å, 5 μm, 2.1 × 100 mm), the product was eluted by gradient as follows: Buffer D (0.1% (v/v) formic acid and 1% (v/v) acetonitrile) for 1 min, then 5% to 95% Buffer E (0.1% (v/v) formic acid in acetonitrile) in 3 min, then 95% Buffer E for 3.5 min. The flow rate was 400 μl min$^{-1}$. The mass spectrometer was operated in negative mode. The source voltage, the curtain gas and the source temperature were set to 4,500 V, 30 psi and 350 °C, respectively. The full scan was performed upon $m/z$ ranging from 120 to 1,200 Da.

## MST assay

The MST assay was performed as described[64]. The binding affinities of GDN-purified WT-Rv3806c or mutants were measured using a Monolith NT.115 (Nanotemper Technologies). The His-tagged protein was labelled with RED fluorescent dye NT-647 according to the manufacturer's procedure. For each assay, the labelled protein at 200 nM was incubated with the same volume of unlabelled ligands at 16 different concentrations in the Buffer C at room temperature for 5 min. Assays of all the mutants were added with 2.5 mM MgCl$_2$. The samples were then loaded into capillaries (NanoTemper Technologies) and measured at 25 °C by using 40% Light Emitting Diodes and medium MST power.

Binding affinities of PRPP with the WT-Rv3806c and mutants were measured under the same parameter. Each assay was repeated three to five times. $K_d$ values were calculated using the MO. Affinity Analysis v.2.2.4 software. All the final plots were made using GraphPad Prism 9.0.

## Trp fluorescence measurements (NanoDSF assay)

Four micromolar GDN-purified Rv3806c protein was incubated with or without MgCl$_2$/PRPP/ethylenediaminetetraacetic acid. Samples were measured by Prometheus NT.48 (NanoTemper) using Prometheus NT.48 Series nanoDSF Grade Standard Capillaries through 15% excitation light. Samples were measured in temperature range 20–95 °C with temperature slope of 1 °C min$^{-1}$ (ref. [65]). The final plots were made using GraphPad Prism 9.0.

## Structure-guided disulfide cross-bridge experiments

To examine structure-based disulfide bridge studies of Rv3806c on *Msm* membrane, cell membranes of *Msm* overexpressing WT-Rv3806c and the T216C/D267C mutant were isolated. Protein loading buffer containing SDS was subsequently added to the membranes to a final concentration of 2% (w/w). The oligomeric status of Rv3806c was identified by western blot using anti-Flag antibody. Reducing agent dithiothreitol was added at room temperature as control.

## Microfluidic modulation spectroscopy

All samples were analysed on an AQS3pro MMS production system (RedShiftBio) using 5 psi backing pressure and 1 Hz modulation with analysis at 29 discrete wavenumbers across the Amide I band (1,589–1,712 cm$^{-1}$). Two replicates were collected for samples at 0.33 mg ml$^{-1}$ in GDN buffer. The delta software was used for sample analysis and spectral processing. The raw differential absorbance spectra were analysed using a 0.63 nominal displacement factor and fit over the full range relative to the model protein spectrum to generate the absolute spectra. The model protein spectrum was obtained by analysing 0.5 mg ml$^{-1}$ WT-Rv3806c protein without spectral fitting. Savitsky–Golay smoothing was applied to the second derivative plot for all samples using a window of 19 wavenumbers[66]. To calculate the percent similarity, the area of overlap was used on the baseline-subtracted second derivative spectrum for each sample using the 'Rubberband' baselining method, which fixes both ends of the second derivative spectrum to the baseline[67]. Higher-order structure structural elements were calculated using the same baseline-corrected plot and Gaussian curve fit settings with nine custom curves based on previous secondary structural infrared peak assignments by Done et al.[68] and Heimburg et al.[69]. Additionally, the higher-order structure bins were set to display the following secondary structural elements in the bar graph: alpha-helix (alpha), beta-sheet (beta), unordered (unord), turn and coiled coil.

## Reporting summary

Further information on research design is available in the Nature Portfolio Reporting Summary linked to this article.

## Data availability

Atomic coordinates of the Rv3806c models were deposited in the RCSB PDB under accession number 8J8K for DP-bound state and 8J8J for PRPP-bound state. The 3D cryo-EM maps were deposited in the Electron Microscopy Data Bank (EMDB) under accession numbers EMD-36072 for DP-bound Rv3806c trimer and 36071 for PRPP-bound Rv3806c trimer. The PDB entries 4TQ5, 4OD5, 8DJM and 1L1R were used for structure comparison in this study. Source data are provided with this paper.

## Code availability

No custom code was used in the analysis of data presented in this manuscript.

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

## Acknowledgements

We thank W. Xu from ShanghaiTech University for discussion of the manuscript. We thank L. Wang and Q. Sun from the Bio-Electron Microscopy Facility and J. Chen, W. Zhu and X. Gao from Analytical Chemistry Platform of Shanghai Institute for Advanced Immunochemical Studies (SIAIS) of ShanghaiTech University for their support in EM data collection and mass spectrometry analysis respectively. We thank J. Liu at the Protein Preparation and Characterization Platform of Technology Center for Protein Research, Tsinghua University for providing facility support in MMS analysis. We thank J. Li from Large-scale Protein Preparation System at the National Facility for Protein Science in Shanghai (NFPS) for technical support. This work was supported by grants from National Natural Science Foundation of China (grant no. 32394010 to Z.R., and grant no. 32394011 to L.Z.), R&D Program of Guangzhou Laboratory (SRPG22-003) to Z.R and L.Z., Shanghai Frontiers Science Center for Biomacromolecules and Precision Medicine, ShanghaiTech University, Medical Research Council UK (MR/S000542/1 and MR/R001154/1) to G.S.B.

## Author contributions

L.Z. and Z.R. conceived the project. L.Z., Z.R. and G.S.B. designed the experiment. S.G. and F.W. performed molecular cloning, protein expression and purification of the WT and mutant Rv3806c. Cryo-EM sample preparation, data collection and data processing were carried out by F.W. and S.G. L.Z. and F.W. analysed the EM data and built and refined the model. S.S.G. and S.M.B. determined the enzymatic activity of WT and mutant Rv3806c under guidance of G.S.B. S.G. performed MST and thermostability experiments. F.W. performed mass spectrometry experiment. All authors analysed and discussed the results. L.Z., G.S.B., F.W. and S.G. wrote the paper with help from all authors.

## Competing interests

The authors declare no competing interests.

## Additional information

**Extended data** is available for this paper at https://doi.org/10.1038/s41564-024-01643-8.

**Correspondence and requests for materials** should be addressed to Gurdyal S. Besra, Zihe Rao or Lu Zhang.

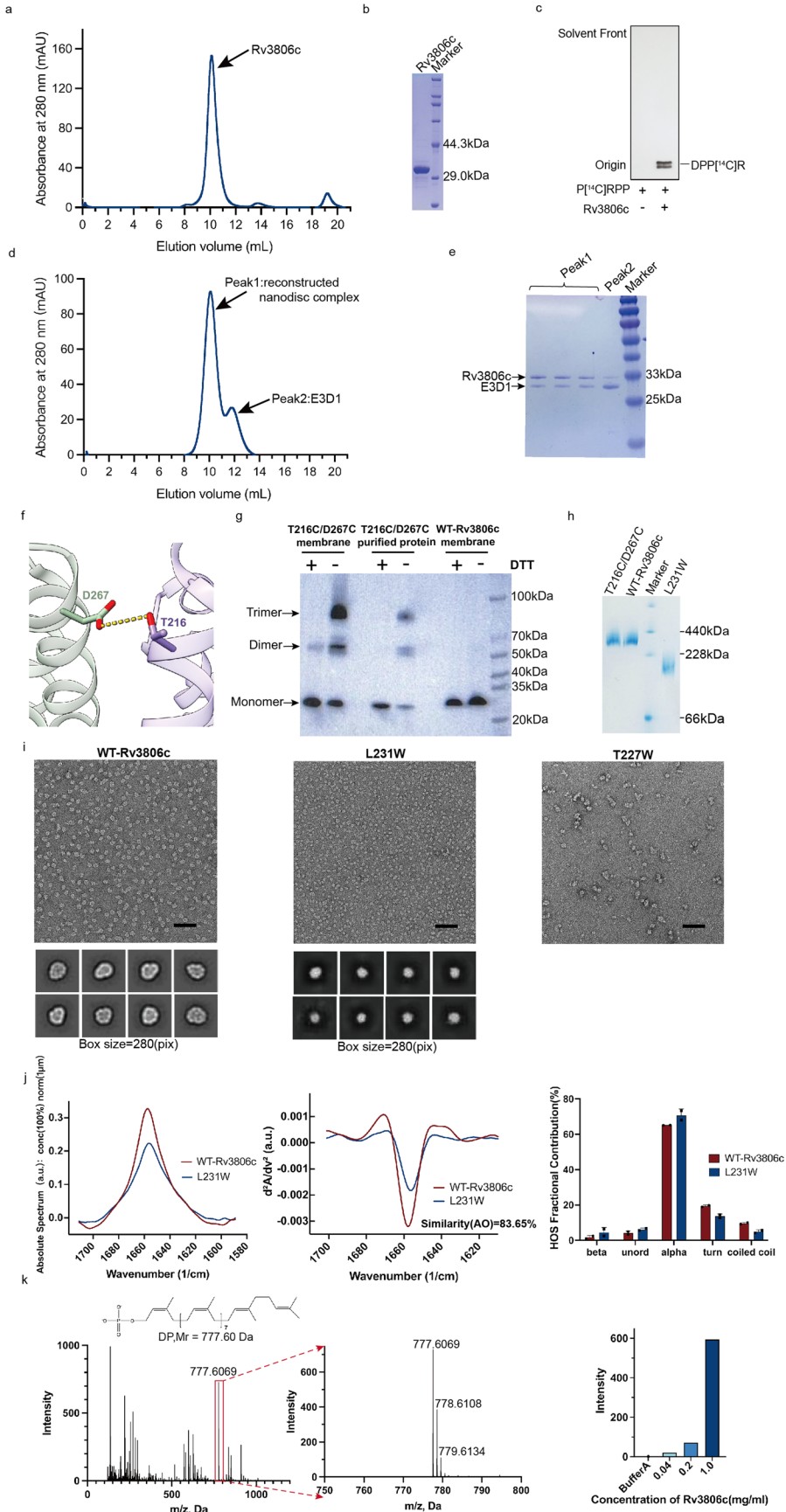

**Extended Data Fig. 1 | See next page for caption.**

**Extended Data Fig. 1 | Protein purification, biochemical characterization and EM analysis. a**, Size–exclusion chromatography for Rv3806c purified with GDN. **b**, The peak fraction was shown on SDS-PAGE with molecular markers. Rv3806c has a predicted molecular weight of 32.6 kDa. **c**, Radiometric-TLC analysis of PRTase activity using radiolabeled P[$^{14}$C]RPP. The DPP[$^{14}$C]R product catalyzed by Rv3806c is shown resolved from [$^{14}$C]ribose phosphate at the origin which was generated in situ through chromatography by TLC as shown (see Source Data Fig. 1). **d**, Representative size-exclusion chromatography trace of Rv3806c reconstituted into MSP-1E3D1 nanodisc. Fractions corresponding to the peak1 were used for cryo-EM analysis. **e**, The peak fractions were shown on SDS-PAGE with molecular markers. **f**, D267 and T216 which mediate the protomer-protomer interface at the cytoplasmic side were selected for double cysteine crossbridge. **g**, Western blot analysis of cross-linking in the absence and presence of detergent, using anti-FLAG antibody. Lane 1-2, *Msm* cell membranes of T216C-D267C mutant; lane 3-4, T216C-D267C mutant purified in GDN; lane 5-6, *Msm* cell membranes of WT-Rv3806c as control; lane 7: molecular weight marker. All samples were analyzed with or without 10 mM reducing agent DTT. The signal of the dimer band in lane 1,2 and 4 may be due to insufficient disulfide bond formation between only two protomers. **h**, Blue Native PAGE analysis of the WT-Rv3806c and mutant Rv3806c samples purified in GDN, the T216C-D267C mutant appears as a trimer as same as the WT-Rv3806c, whereas the molecular

weight of the L231W mutant is lower than the WT-Rv3806c. **i**, The representative micrograph of negative stained Rv3806c samples purified in GDN solution. (left) WT-Rv3806c shows homogeneous particles. Reference-free 2D class averages of particles revealing a 3-fold symmetric feature. (middle) The L231W mutant shows smaller particles as well as smaller 2D class averages of particles of this mutant compared with the WT-Rv3806c sample. (right) The T227W mutant appears as aggregates. Scale bar: 50 nm. **j**, MMS data shows both WT-Rv3806c and L231W were folded in solution with an alpha-helix dominated structure. The absolute spectra (left) and second derivative spectra (middle) show the main peaks around 1657-1658 cm-1 which is a signature peak for alpha-helix structure in the Amide I band. (right) Higher Order Structure (HOS) fractional contributions of the L231W mutant and WT-Rv3806c protein. The percentage secondary structural elements were calculated by Gaussian curve fitting on the baseline-corrected second derivative spectra (see Supplementary Table 3). Data presented are mean +SD calculated from two independent experiments for both WT and L231W mutant Rv3806c. **k**, (left) Full scan mode mass spectrometry analysis of solvent extracted DP from Rv3806c purified in GND solution, the ion of m/z 777.6, which corresponds to the peak of DP, is indicated, (middle) isotopic distributions of DP are indicated within m/z ranging from 750 to 800 Da. (right) MS intensity of DP was measured in a dose dependent manner to protein concentration.

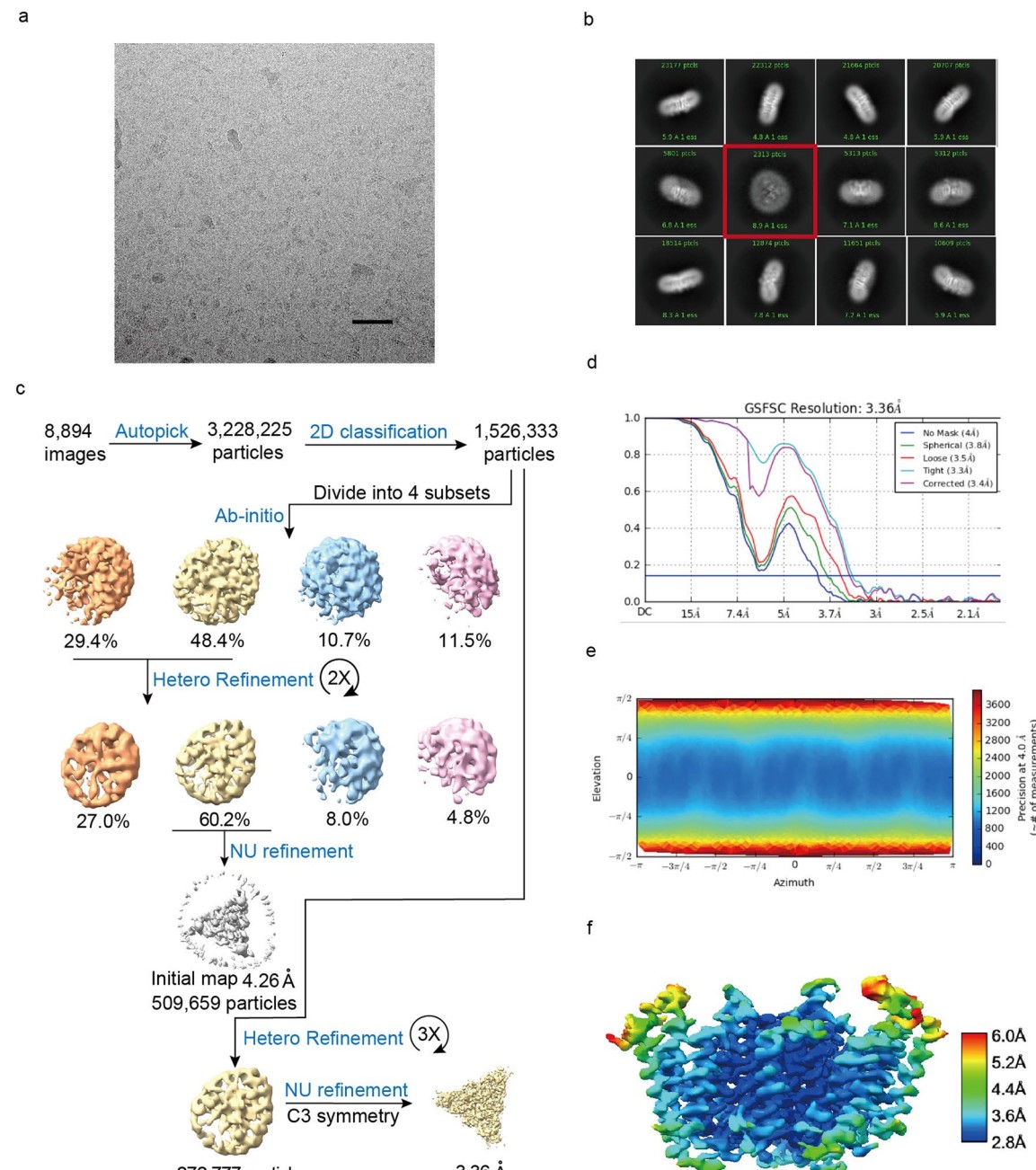

**Extended Data Fig. 2 | Data processing of DP-bound Rv3806c and refined models. a**, Representative motion-corrected cryo-EM micrograph (out of 8,894 micrographs). Scale bar: 50 nm. **b**, Selected reference-free 2D class averages. **c**, Flow chart for the processing of cryo-EM data. **d**, Gold-standard fourier correlation curves of 3D reconstructions. **e**, Posterior precision directional distributions of all particles used in the final 3D reconstruction reported by cryoSPARC. **f**, The density map colored according to the local resolution estimation using cryoSPARC.

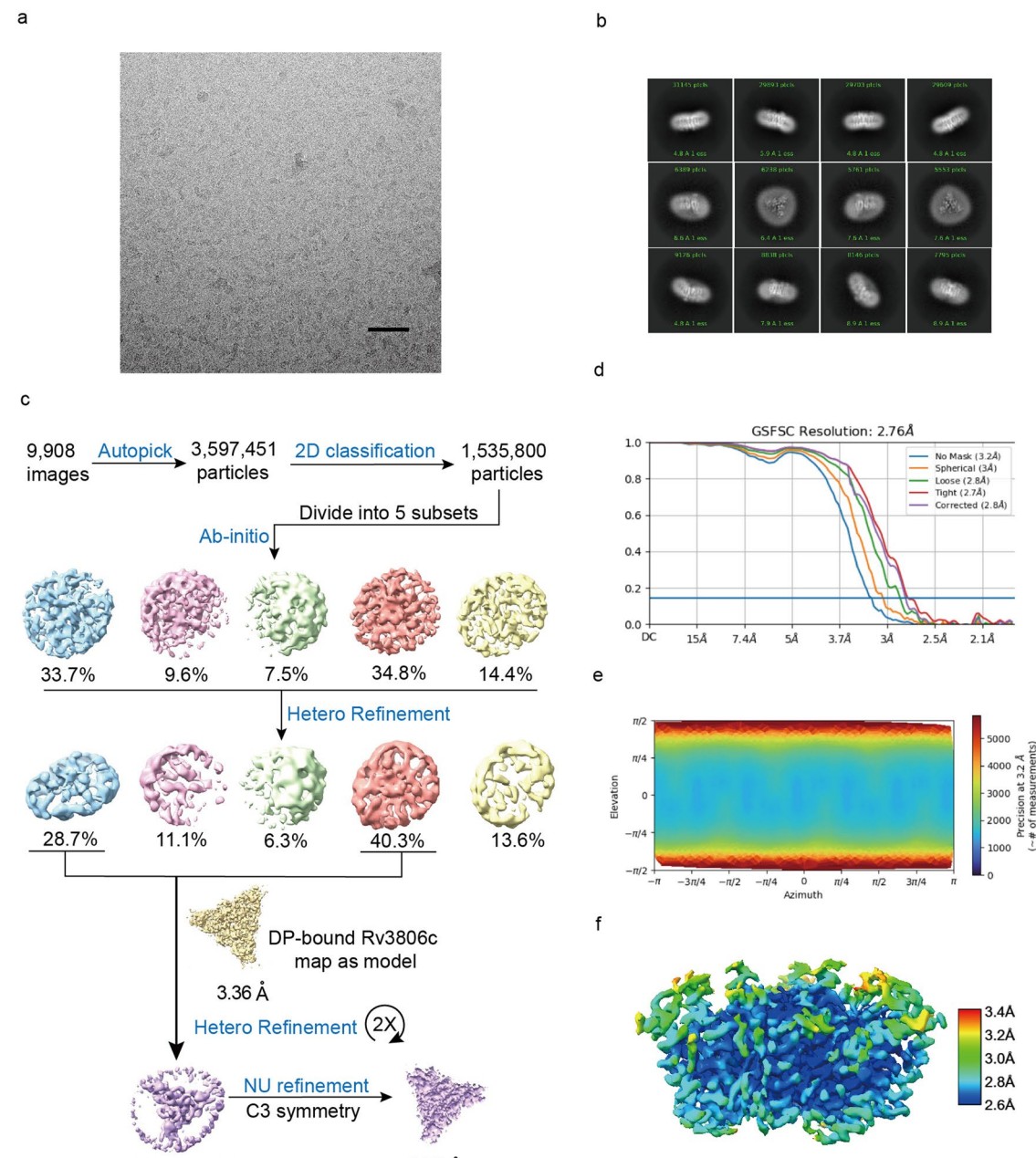

**Extended Data Fig. 3 | Data processing of PRPP-bound Rv3806c and refined models. a**, Representative motion-corrected cryo-EM micrograph (out of 9,908 micrographs). Scale bar: 50 nm. **b**, Selected reference-free 2D class averages. **c**, Flow chart for the processing of cryo-EM data. **d**, Gold-standard fourier correlation curves of 3D reconstructions. **e**, Posterior precision directional distributions of all particles used in the final 3D reconstruction reported by cryoSPARC. **f**, The density map colored according to the local resolution estimation using cryoSPARC.

**Extended Data Fig. 4 | Electron densities for the proteins and ligands from DP-bound and PRPP-bound Rv3806c. a**, The cryo-EM map (threshold 0.35) of TM1-9 and DP from the DP-bound Rv3806c map. **b**, The cryo-EM map (threshold 0.49) of TM1-9, POPG, PRPP and Mg$^{2+}$ from the PRPP-bound map.

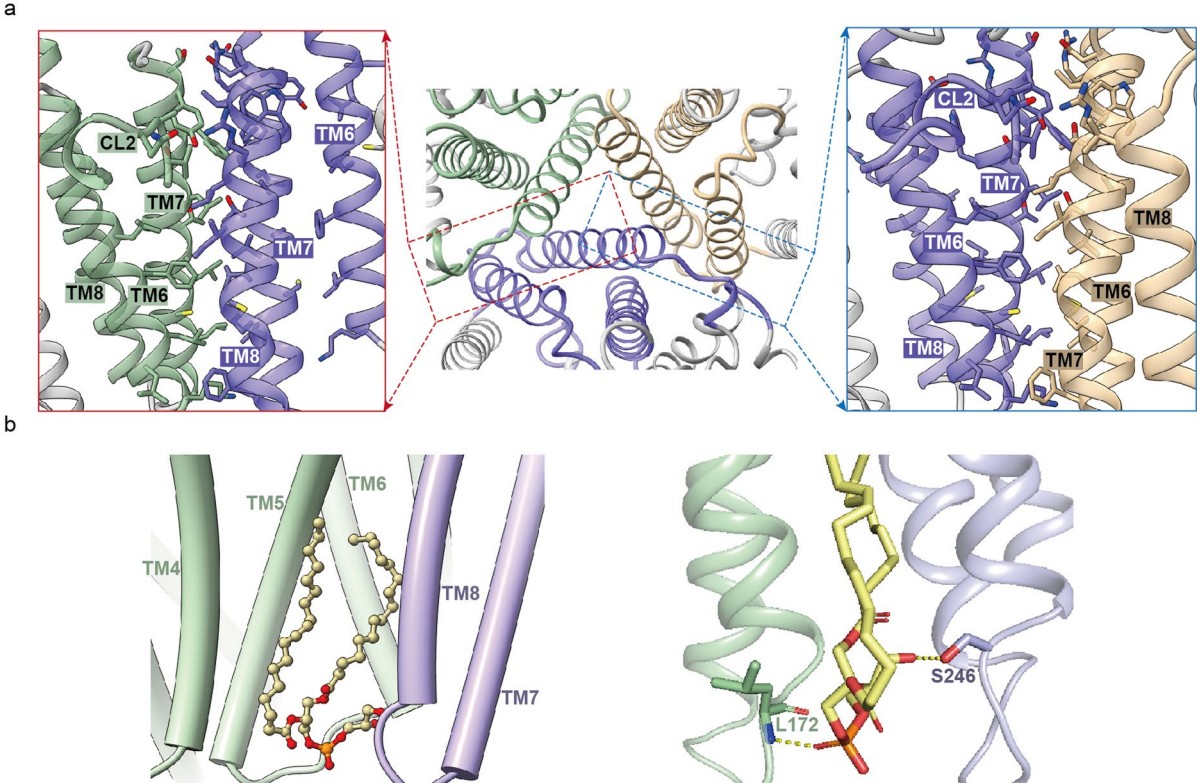

**Extended Data Fig. 5 | Interactions of the protomer-protomer interface and POPG binding sites. a**. The protomer-protomer interfaces in the (middle) trimeric Rv3806c are shown by two dashed boxes. The left and right zoomed-in inlets show residues on TM 6/7/8 participating in the protomer-protomer interaction. TM helices are represented as ribbons and residues are shown as sticks. **b**. (left) POPG binding in the periplasmic interface formed by TM 4-6 of one protomer (green) and TM 7-8 of the other (purple). POPG is shown as (left) balls and sticks and (right) sticks, polar binding sites of POPG are shown as sticks.

a

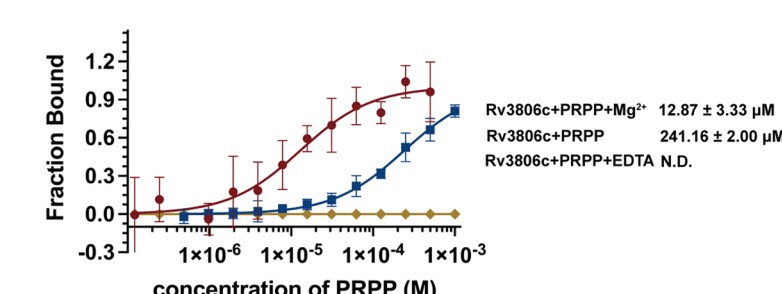

Rv3806c+PRPP+Mg²⁺    12.87 ± 3.33 µM
Rv3806c+PRPP          241.16 ± 2.00 µM
Rv3806c+PRPP+EDTA     N.D.

b

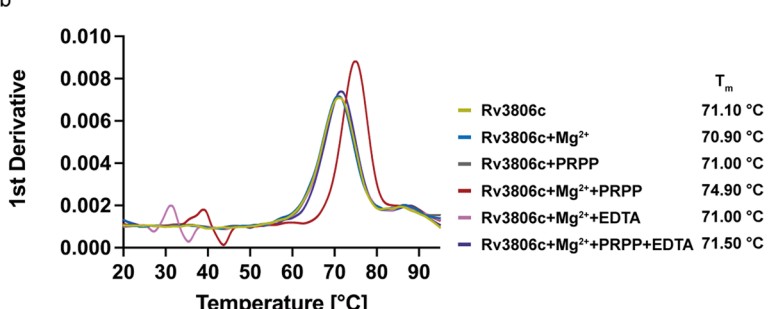

**Extended Data Fig. 6 | Binding affinity analyzed by MST assay and the thermal stability of Rv3806c. a**. Binding affinity of PRPP to Rv3806c in the presence or absence of Mg²⁺ measured by the MST assay. Binding affinity dramatically decreased in the absence of Mg²⁺ and that no binding was detected in the presence of the metal chelator EDTA. The $K_d$ values are provided; data are representative mean values ± SDs, calculated from five independent experiments. **b**. NanoDSF analysis of the thermal stability of purified Rv3806c in the presence or absence of Mg²⁺, PRPP and EDTA. Rv3806c shows the highest thermostability in the presence of PRPP-Mg²⁺ than adding PRPP or Mg²⁺ alone. The melting temperature (Tm) values are provided.

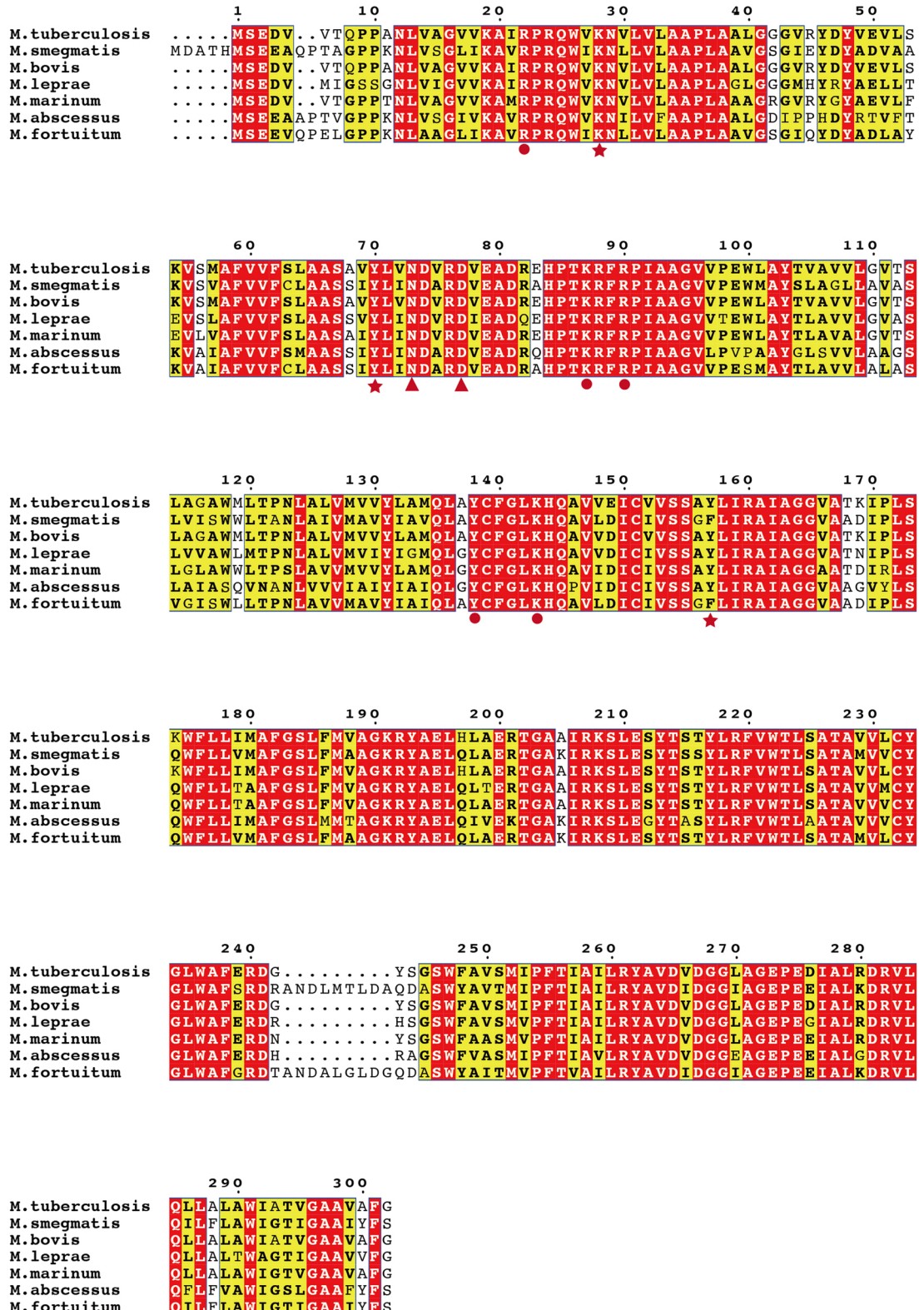

**Extended Data Fig. 7 | Sequence alignment of Rv3806c homologs from *Mycobacteria*.** Sequence alignment of Rv3806c from selected *Mycobacteria*. Mg$^{2+}$ binding sites, phosphoribose binding sites and PP$_i$ binding sites in Rv3806c structure are highlight below the aligned sequences as red triangles, pentagons and circles, respectively.

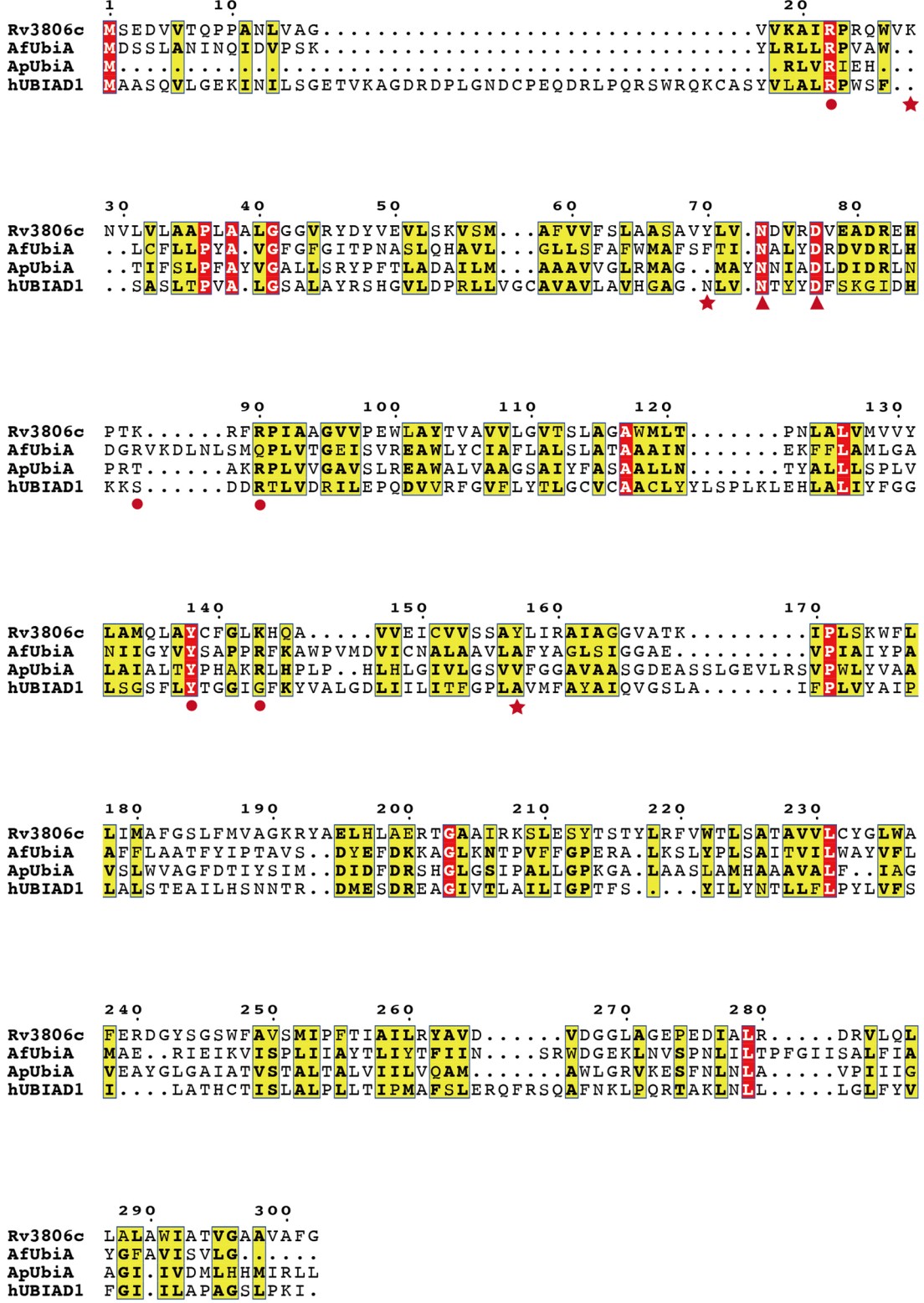

**Extended Data Fig. 8 | Sequence alignment of Rv3806c and UbiA superfamily.** Sequence alignment of *Mycobacteria tuberculosis* Rv3806c, *Archaeoglobus fulgidus* UbiA, *Aeropyrum pernix* UbiA and Human UBIAD1. Mg²⁺ binding sites, phosphoribose binding sites and PPᵢ binding sites in Rv3806c structure are highlight below the aligned sequences as red triangles, pentagons and circles, respectively.

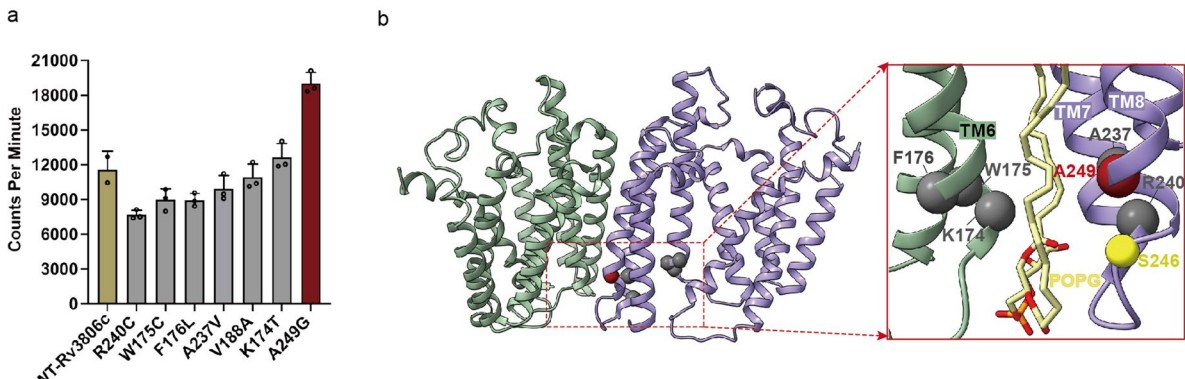

**Extended Data Fig. 9 | Mapping of clinical ethambutol-resistant Rv3806c mutations and enzymatic analysis. a**. PRTase enzymatic activity analysis of clinical ethambutol resistant Rv3806c mutations. Data presented are mean +SD calculated from three independent experiments for the mutant Rv3806c proteins, and for the WT-Rv3806c proteins from two independent experiments (see Source Data Extended Data Fig. 9). **b**. (Left) mapping of the clinical ethambutol resistant mutation sites of Rv3806c on the structure. Mutation sites are shown as spheres. The red spheres are the mutation sites with enhanced enzymatic activity compared to WT-Rv3806c. The dashed box indicates the region where mutation sites are clustered. (Right) zoomed-in inlets show details of the mutation sites around the POPG binding pocket. The yellow spheres represent the binding sites of POPG (see also Extended Data Fig. 5b right).

# Reporting Summary

## Statistics

For all statistical analyses, confirm that the following items are present in the figure legend, table legend, main text, or Methods section.

| n/a | Confirmed | |
|---|---|---|
| ☐ | ☒ | The exact sample size ($n$) for each experimental group/condition, given as a discrete number and unit of measurement |
| ☐ | ☒ | A statement on whether measurements were taken from distinct samples or whether the same sample was measured repeatedly |
| ☒ | ☐ | The statistical test(s) used AND whether they are one- or two-sided<br>*Only common tests should be described solely by name; describe more complex techniques in the Methods section.* |
| ☒ | ☐ | A description of all covariates tested |
| ☒ | ☐ | A description of any assumptions or corrections, such as tests of normality and adjustment for multiple comparisons |
| ☐ | ☒ | A full description of the statistical parameters including central tendency (e.g. means) or other basic estimates (e.g. regression coefficient) AND variation (e.g. standard deviation) or associated estimates of uncertainty (e.g. confidence intervals) |
| ☒ | ☐ | For null hypothesis testing, the test statistic (e.g. $F$, $t$, $r$) with confidence intervals, effect sizes, degrees of freedom and $P$ value noted<br>*Give P values as exact values whenever suitable.* |
| ☒ | ☐ | For Bayesian analysis, information on the choice of priors and Markov chain Monte Carlo settings |
| ☒ | ☐ | For hierarchical and complex designs, identification of the appropriate level for tests and full reporting of outcomes |
| ☒ | ☐ | Estimates of effect sizes (e.g. Cohen's $d$, Pearson's $r$), indicating how they were calculated |
| | | *Our web collection on statistics for biologists contains articles on many of the points above.* |

## Software and code

Policy information about availability of computer code

| Data collection | EPU2.12 |
|---|---|
| Data analysis | MotionCor2 1.2.1, cryoSPRAC 4.0.1, Phenix 1.17, COOT 0.9.8, Pymol 2.1,UCSF Chimera X 1.4, GraphPad prism 9.0. |

For manuscripts utilizing custom algorithms or software that are central to the research but not yet described in published literature, software must be made available to editors and reviewers. We strongly encourage code deposition in a community repository (e.g. GitHub). See the Nature Portfolio guidelines for submitting code & software for further information.

## Data

Policy information about availability of data

All manuscripts must include a data availability statement. This statement should provide the following information, where applicable:

- Accession codes, unique identifiers, or web links for publicly available datasets
- A description of any restrictions on data availability
- For clinical datasets or third party data, please ensure that the statement adheres to our policy

The EM density maps generated in this study have been deposited in the EMDB under accession codes EMD-36072 ( DP-bound Rv3806c), EMD-36071 (PRPP-bound Rv3806c). Atomic coordinates have been deposited in the PDB under the accession codes 8J8K (DP-bound Rv3806c), and 8J8J (PRPP-bound Rv3806c).

# Research involving human participants, their data, or biological material

Policy information about studies with [human participants or human data](). See also policy information about [sex, gender (identity/presentation), and sexual orientation]() and [race, ethnicity and racism]().

| | |
|---|---|
| Reporting on sex and gender | not applied |
| Reporting on race, ethnicity, or other socially relevant groupings | not applied |
| Population characteristics | not applied |
| Recruitment | not applied |
| Ethics oversight | not applied |

Note that full information on the approval of the study protocol must also be provided in the manuscript.

# Field-specific reporting

Please select the one below that is the best fit for your research. If you are not sure, read the appropriate sections before making your selection.

☒ Life sciences          ☐ Behavioural & social sciences          ☐ Ecological, evolutionary & environmental sciences

For a reference copy of the document with all sections, see [nature.com/documents/nr-reporting-summary-flat.pdf]()

# Life sciences study design

All studies must disclose on these points even when the disclosure is negative.

| | |
|---|---|
| Sample size | sample sizes were not predetermined and all available data was processed. For cryo-EM, two datasets with 8894 micrographs (DP bound) and 9908 micrographs (PRPP bound) were collected and processed. Data of the DP-bound and PRPP-bound Rv3806c samples were processed to a resolution of 3.36Å and 2.76Å, both of which were sufficient for interpretation of the experimental data and to build an atomic model. |
| Data exclusions | No data exclusions in experimental groups except being cryo EM data processing, we discarded the "junk" particles or classes of particles that did not yield a useful 3D reconstruction. This followed the standard cryo-EM processing procedure and commonly accepted in the cryo-EM single particle analysis field. |
| Replication | Protein purification of WT-Rv3806c and mutant-Rv3806c, SDS-PAGE, BN-PAGE, West-Blot were repeated in at least 3 independent experiments. Scintillation counting of PRTase activity assay for all WT and mutant Rv3806c was repeated in 2-3 independent experiments. Binding affinity assays were repeated in 3-5 independent experiments.Negative stain of WT-Rv3806c and mutant-Rv3806c was repeated 3 in independent experiments. Cryo-EM sample preparation and data processing were repeated twice. The TLC-autoradiogram for the RPTase activity was performed once. Microfluidic modulation spectroscopy assay was repeated twice.All attempts at replication were successful. |
| Randomization | Random allocation with regard to covariate is not applicable to the experiments carried out. Experiments in this study do not necessitate randomization because they are not influenced by covariates during the sample allocation common in other research methods. |
| Blinding | Blinding was not required for this study because nor subjective allocation was involved, neither did data processing rely on subjective assignments. Because the biochemical/biophysical data was visualized by scintillation counting, or absorbance and quantified using standard software, which did not require subjective analysis. And structural data analyzed by standard software packages which didnot require subjective judgment. |

# Reporting for specific materials, systems and methods

We require information from authors about some types of materials, experimental systems and methods used in many studies. Here, indicate whether each material, system or method listed is relevant to your study. If you are not sure if a list item applies to your research, read the appropriate section before selecting a response.

## Materials & experimental systems

| n/a | Involved in the study |
|---|---|
| ☐ | ☒ Antibodies |
| ☒ | ☐ Eukaryotic cell lines |
| ☒ | ☐ Palaeontology and archaeology |
| ☒ | ☐ Animals and other organisms |
| ☒ | ☐ Clinical data |
| ☒ | ☐ Dual use research of concern |
| ☒ | ☐ Plants |

## Methods

| n/a | Involved in the study |
|---|---|
| ☒ | ☐ ChIP-seq |
| ☒ | ☐ Flow cytometry |
| ☒ | ☐ MRI-based neuroimaging |

## Antibodies

| | |
|---|---|
| Antibodies used | HPR-conjugated mouse anti DDDDK-Tag(1:5000,ABClone,AE024,China) was used for the Western Blot analysis of WT and mutant Rv3806c. |
| Validation | Antibody was exclusively used for western blotting. Validation of the antibody can be found on the website (https://abclonal.com.cn/catalog/AE024). |

