## [Peer Review File · Nature Microbiology]

Peer Review Information

Journal: Nature Microbiology

Manuscript Title: Structural analysis of phosphoribosyltransferase-mediated cell wall precursor synthesis in *Mycobacterium tuberculosis*

Corresponding author name(s): Zihe Rao

Reviewer Comments & Decisions:Decision Letter, initial version:

Message: 28th July 2023

Dear Professor Rao,

Thank you for your patience while your manuscript "Structure and mechanism of a membrane-bound phosphoribosyltransferase central to *M. tuberculosis* cell wall biosynthesis" was under peer-review at Nature Microbiology. It has now been seen by 3 referees, whose expertise and comments you will find at the end of this email. Although they find your work of some potential interest, they have raised a number of concerns that will need to be addressed before we can consider publication of the work in Nature Microbiology.

In particular, you will see that Referee #1 requested improved data to support the conclusions that the T227W and L231W mutants of Rv3068c do not form trimers, and to show that these proteins are not denatured. In addition, there was a request for better resolution in the mass spectrometry data supporting detection of DP, plus a number of additional requests to discuss alternative interpretations of data or tone down some conclusions. Furthermore, after further discussions with the referees, they highlighted similarities in the normalised PRTase activity data for WT Rv3068c across several figure panels, plus a discrepancy between the number of replicates shown in the plot and that reported in the figure legend. It is essential that this potential issue be addressed, and we would need a revised manuscript to provide raw data, including copies of the source autoradiographs, to clarify that these control data come from independent experiments. These are critical points which must be addressed for us to consider a revised manuscript, alongside the rest of the comments provided in the referees' reports, which are clear and should be straightforward to address.

Should further experimental data allow you to address these criticisms, we would be happy to look at a revised manuscript.

3Please include a data availability statement as a separate section after Methods but before references, under the heading "Data Availability". This section should inform readers about the availability of the data used to support the conclusions of your study. This information includes accession codes to public repositories (data banks for protein, DNA or RNA sequences, microarray, proteomics data etc...), references to source data published alongside the paper, unique identifiers such as URLs to data repository entries, or data set DOIs, and any other statement about data availability. At a minimum, you should include the following statement: "The data that support the findings of this study are available from the corresponding author upon request", mentioning any restrictions on availability. If DOIs are provided, we also strongly encourage including these in the Reference list (authors, title, publisher (repository name), identifier, year). For more guidance on how to write this section please see: <http://www.nature.com/authors/policies/data/data-availability-statements-data-citations.pdf>

* If you have not done so already we suggest that you begin to revise your manuscript so that it conforms to our Article format instructions at <http://www.nature.com/nmicrobiol/info/final-submission>. Refer also to any guidelines provided in this letter.

When submitting the revised version of your manuscript, please pay close attention to our [href="https://www.nature.com/nature-portfolio/editorial-policies/image-integrity">Digital Image Integrity Guidelines](https://www.nature.com/nature-portfolio/editorial-policies/image-integrity). and to the following points below:

Note: This url links to your confidential homepage and associated information about manuscripts you may have submitted or be reviewing for us. If you wish to forward this e-mail to co-authors, please delete this link to your homepage first.

Nature Microbiology is committed to improving transparency in authorship. As part of our efforts in this direction, we are now requesting that all authors identified as 'corresponding author' on published papers create and link their Open Researcher and Contributor Identifier (ORCID) with their account on the Manuscript Tracking System (MTS), prior to acceptance. This applies to primary research papers only. ORCID helps the scientific community achieve unambiguous attribution of all scholarly contributions. You can create and link your ORCID from the home page of the MTS by clicking on 'Modify my Springer Nature account'. For more information please visit www.springernature.com/orcid.

If you wish to submit a suitably revised manuscript we would hope to receive it within 6 months. If you cannot send it within this time, please let us know. We will be happy to consider your revision, even if a similar study has been accepted for publication at Nature Microbiology or published elsewhere (up to a maximum of 6 months).

Reviewer Expertise:

Referee #1: CryoET

Referee #2: Mycobacterial cell wall and membrane biology

Reviewer Comments:

Reviewer #1 (Remarks to the Author):

The paper describes the structure of the membrane-bound Rv3806c (spans the cytoplasm and periplasm), a phosphoribosyl transferase, whereby ligand PRPP targets membrane bound ligand decaprenol phosphate (DP), to generate the product ligand DPPR and release of pyrophosphate. RV3806c is composed of 9 transmembrane (TM) helices divided into two helical bundles that are connected through periplasmic and cytoplasmic loops. Notably, Rv3806c adopts a trimeric alignment stabilized by protomer-protomer interfaces and lipids, with the trimeric state critical in the mediation of phosphoribose transfer. The authors solved the structure of Rv3806c bound to DP and could observe the phosphate and some of the prenyl linkage positioned in a cavity within the TM segment connected to an extramembrane cleft opening to the cytoplasm. The DP-binding cavity in Rv3806c

5is distinct from its counterpart in UbiA superfamily enzymes. They also solved the structure of Rv3806c bound to endogenous PRPP-Mg ligand, wherein the phosphate group is buried in the membrane. Key residues contacting both bound DP and bound PRPP were mutagenized to monitor their impact on catalysis. Based on the positioning of DP and PRPP, an inverted catalytic mechanism was proposed for formation of product DPPR.

Overall, this was a challenging structural project brought to successful conclusion, which was complemented by mutagenesis studies to provide insights into the catalytic mechanism. I recommend publication in its current form but would encourage the authors in future studies to try and solve the structure of Rv3806c bound to product DPPR, towards a fuller understanding of the catalytic cycle.

Reviewer #2 (Remarks to the Author):

This study reports the cryo EM structure of Rv3806c, one of the critical enzymes for the synthesis of arabinose donor in mycobacteria. The detailed structural and biochemical analysis provides exciting new insights on how this unusual membrane-embedded enzyme utilizes PRPP to produce DPPR. In particular, cryo-EM structures of the enzyme with bound substrates provide interesting mechanistic understanding of this very important enzyme. Another notable achievement is the establishment of the enzymatic assay system for this membrane-embedded enzyme. Overall, the study was rigorously done, and the results are generally supportive of the author's conclusions. However, we have several major questions and thoughts that the authors can clarify further.

Major comments

We are not convinced of the quality of the data presented in Fig. 2e. The native SDS-PAGE gel is without standards, and it is not even clear if these two smeary bands are different in mobility. Can the authors run both L231W and WT together in a single lane on the native PAGE to show that they can be separated in a single lane? SEC is supposed to be more quantitative than native PAGE, but the authors did not show the molecular weight markers. The authors claim that T227W came out in the void volume, but they did not provide any evidence for this. There is also no evidence for them to claim that T227W was denatured. The suggestion that L231W is not forming a complex equivalent to the WT trimer is acknowledged based on the SEC data. However, the data is not sufficient to claim that L231W is properly folded. Can the authors do cryo-EM of this mutant, or at least CD or some other structural analysis to show that it's not denatured? Overall, Fig. 2e is of poor quality, and needs to be revised.

In Extended Figure 1e, why don't we observe isotopic distributions? The resolution of the spectrum in the figure is very poor, and it is not very convincing. Can authors do fragmentation to make a more convincing case? What does the blue arrow on the y-axis mean?

The higher turnover rate of K174T claimed in Extended Fig 9a is not convincing. Please provide a statistical analysis to substantiate this claim. In addition, we would like the authors to discuss an alternative possibility. It is equally possible

that the periplasmic side of the enzyme is used for feedback inhibition by DPA. The mutations make the enzyme unresponsive to DPA, making it constitutively active. In such a scenario, you don't have to assume enhanced activities from the mutant enzymes. Instead, the authors should observe inhibition of the WT enzyme, but not mutant enzymes, by DPA in their enzyme assay. We don't think that the authors have to do this experiment, but the authors should discuss such an alternative possibility (which we think is actually more feasible and exciting).

By looking at Fig. 5c, we are struck by the enzyme's inner core appearing to have a channel penetrating through the membrane all the way from cytoplasmic to periplasmic spaces. Is this true? If there is a channel in this enzyme, does a similar channel exist in other UbiA homologs? Is it possible that DP in the periplasmic side may also be acquired by this enzyme, possibly through this channel? If so, this enzyme can effectively function as a DP flippase, which we feel is an exciting possibility. Can authors clarify the structure in this regard and discuss it more in the manuscript?

We are not convinced with the proposed steps of the enzymatic reaction proposed in Fig. 5c. The authors propose that DP comes in first based on the fact that the enzyme was purified with a bound DP. However, when they incubated the enzyme with PRPP, they no longer found DP in the enzyme structure. Their reasoning in Line 233-235 is reasonable, but that means that the enzyme can exist as a PRPP-bound form without DP. Based on these observations, it seems like the enzyme can accept either substrate first. To us, there appears to be no convincing evidence for the authors to suggest that DP binds to the enzyme before PRPP does.

Finally, there is no discussion on why this enzyme must be a trimer. Can the authors provide any insights into this particular finding? How might this arrangement aid in the mechanism of catalysis?

Minor comments

Line 28: "Kingdom" is an outdated concept. "Domain" is more appropriate.

Line 34: Why is it unexpected that PRPP is used in the biosynthesis of AG and LAM? This was not clear to us.

Line 68: We don't think GDN detergent is the key method information for the enzyme purification. The authors did not use GDN to purify Rv3806c. Rather, they used anti-FLAG resin to pull down FLAG-tagged Rv3806c. Perhaps, it is more appropriate to say that GDN was used to solubilize Rv3806c.

Line 69: Please provide a citation for your established cell-free PRTase assay.

Line 77: It is confusing to state that DP was clearly resolved because only a portion of DP is visible. The authors do clarify this point later in the manuscript. However, without knowing what the authors meant, this statement was confusing and misleading.

Line 140: Fig. 3e – do you mean Fig. 3d? Note that the figure legend is also

messed up.

Line 143: There is no Fig. 3f.

Line 149: It should be Fig. 3b. Review your manuscript more carefully before submission.

Line 166: Please indicate Extended Figure 6a to support this sentence.

Line 170: Please indicate Extended Figure 6b specifically.

Line 191: Did you mean to reference Figure 4b? If you really want us to look at Fig. 4c, please indicate Lys28, Tyr70, and Tyr157 in this panel. Also, 3-letter code for tyrosine is "Tyr" and not "Try".

Line 212: POPG was never defined anywhere. Please define.

Line 316: It is inaccurate to state that cell wall biosynthesis in Mtb (as a whole) is catalyzed by Rv3806c.

Line 346: "Normalized PRTase analysis" is not descriptive enough. This line should say "Normalized PRTase activity analysis" or something similar.

Line 353, 354: Can you specify the type of molecular interactions that are indicated by the dashed lines?

Line 354: The bound substrate in the structure contains a PPI moiety, but it is inaccurate to refer to this substrate as PPI as if it were a free pyrophosphate.

Figure 1a. While studies have shown LAM to be surface exposed, LAM's size makes it unlikely to span from the inner membrane through the outer membrane (in fact, the position of the arabinan domain of LAM intercalated with outer membrane lipids as shown here is not supported in the literature). Additionally, AG should be shown to be covalently linked to the outer membrane in this cartoon. It is also a bizarre way of color-coding, like blue "M" referring to mannan. Finally, the plasma membrane is misspelled as "plasm membrane".

Figure 2f. Why are there four dots from WT if it was done in triplicate? The bar graph presumably shows the average, but it does not look like the average of the four dots. What are the data normalized t

Author Rebuttal to Initial comments

Reviewer Expertise:

Referee #1: CryoET

Referee #2: Mycobacterial cell wall and membrane biology

8Reviewer Comments:

Reviewer #1 (Remarks to the Author):

The paper describes the structure of the membrane-bound Rv3806c (spans the cytoplasm and periplasm), a phosphoribosyl transferase, whereby ligand PRPP targets membrane bound ligand decaprenol phosphate (DP), to generate the product ligand DPPR and release of pyrophosphate. Rv3806c is composed of 9 transmembrane (TM) helices divided into two helical bundles that are connected through periplasmic and cytoplasmic loops. Notably, Rv3806c adopts a trimeric alignment stabilized by protomer-protomer interfaces and lipids, with the trimeric state critical in the mediation of phosphoribose transfer. The authors solved the structure of Rv3806c bound to DP and could observe the phosphate and some of the prenyl linkage positioned in a cavity within the TM segment connected to an extramembrane cleft opening to the cytoplasm. The DP-binding cavity in Rv3806c is distinct from its counterpart in UbiA superfamily enzymes. They also solved the structure of Rv3806c bound to endogenous PRPP-Mg ligand, wherein the phosphate group is buried in the membrane. Key residues contacting both bound DP and bound PRPP were mutagenized to monitor their impact on catalysis. Based on the positioning of DP and PRPP, an inverted catalytic mechanism was proposed for formation of product DPPR.

Overall, this was a challenging structural project brought to successful conclusion, which was complemented by mutagenesis studies to provide insights into the catalytic mechanism. I recommend publication in its current form but would encourage the authors in future studies to try and solve the structure of Rv3806c bound to product DPPR, towards a fuller understanding of the catalytic cycle.

Response: We very much appreciated Reviewer #1 for the supportive comments to our manuscript and suggestions on future studies of Rv3806c towards a fuller understanding of the catalytic cycle. The difficulty in solving the structure of Rv3806c with its product DPPR is the challenging chemosynthesis of DPPR which contains a highly hydrophobic decaprenyl tail. We aim to solve the DPPR-bound structure and look forward to the success on chemosynthesis of DPPR .

Reviewer #2 (Remarks to the Author):

This study reports the cryo EM structure of Rv3806c, one of the critical enzymes for the synthesis of arabinose donor in mycobacteria. The detailed structural and biochemical analysis provides exciting new insights on how this unusual membrane-embedded enzyme utilizes PRPP to produce DPPR. In particular, cryo-EM structures of the enzyme with bound substrates provide interesting mechanistic understanding of this very important enzyme. Another notable achievement is the establishment of the enzymatic assay system for this membrane-embedded enzyme. Overall, the study was rigorously done, and the results are generally supportive of the author's conclusions. However, we have several major questions and thoughts that the authors can clarify further.

Response: We appreciated the reviewer #2 for his/her overall supportive comments to our work. By reading the following comments, the authors recognized Reviewer #2 as a real expert in *Mtb* cell wall biology, and the suggestions will greatly help improve the manuscript quality to merit publication on Nature Microbiology.

Major comments

We are not convinced of the quality of the data presented in Fig. 2e. The native SDS-PAGE gel is without standards, and it is not even clear if these two smeary bands are different in mobility. Can the authors run both L231W and WT together in a single lane on the native PAGE to show that they can be separated in a single lane? SEC is supposed to be more quantitative than native PAGE, but the authors did not show the molecular weight markers. The authors claim that T227W came out in the void volume, but they did not provide any evidence for this. There is also no evidence for them to claim that T227W was denatured. The suggestion that L231W is not forming a complex equivalent to the WT trimer is acknowledged based on the SEC data. However, the data is not sufficient to claim that L231W is properly folded. Can the authors do cryo-EM of this mutant, or at least CD or some other structural analysis to show that it's not denatured? Overall, Fig. 2e is of poor quality, and needs to be revised.

Response: Thanks for pointing out the trimeric assembly issue. Based on the Reviewer #2 suggestion, we have

- 1) Improved the quality of the native PAGE to better show the difference between WT band and L231W mutant band, together with the molecular weight standards. To be noted, both the WT and the mutant bands showed much higher molecular

10weights in the native PAGE than theoretical molecular weights (96kD for trimer, 64kD for dimer and 32kD for monomer) . This is due to the fact that membrane protein purified in detergent micelles (GDN in this study) are commonly larger than theoretical molecular weight. The updated BN PAGE data was added as a figure panel in Extended Data Fig. 1h (together with the newly added dimer interface crosslink data to study the oligomeric state of Rv3806c in its native membrane) in the revised manuscript for your

review. Given that the WT and the L231W mutant are well separated in the native PAGE, the two samples were not mixed in the single lane.

Extended Data Fig. 1h

- 2) Labelled the elution volume of WT and mutants samples in the SEC result, it could be observed in the SEC data that T227W was eluted close to the void volume in the same purification buffer as the WT sample. This result has also been

updated in Figure panel 2e in the revised manuscript for your review.

Fig. 2e

- 3) Add negative staining EM analysis of WT and mutant Rv3806c samples purified in GDN solution to study their oligomeric states.

The representative micrograph of negative stained WT sample showing homogeneous particles. Reference-free 2D class averages of particles revealing a 3-fold symmetric feature. The data indicated that Rv3806c forms a trimer in GDN solution, in consistent with the cryo-EM structure of the nanodisc reconstituted Rv3806c sample.

The representative micrograph of negative stained L231W mutant sample showing smaller particles as well as smaller 2D class averages of particles compared to the WT sample, and it is technically difficult to do further cryo-EM data analysis. To be noted, at this resolution it is difficult to distinguish the L231W mutant sample is monomer or dimer, so we concluded that this mutant forms a lower oligomer than the trimeric WT sample.

The representative micrograph of negative stained T227W mutant showing particles that appear as aggregates, in consistent with its behavior in the SEC

analysis whereby it was eluted close to the void volume. Therefore, in the revised manuscript, we concluded that the T227W mutant was improperly folded. The negative staining EM analysis of WT, L231W and L227W mutant Rv3806c was added as a figure panel in the Extended Data Figure 1i in the revised manuscript for your review.

Extended Data Figure 1i

- 4) Add the microfluidic modulation spectroscopy (MMS) analysis to study the secondary structural feature of the L231W mutant. The MMS data shows that both WT and the L231W mutant Rv3806c were properly folded in GDN solution with an alpha-helix dominated structure. The minor difference between the WT and L231W mutant proteins may be due to the detramerization. The MMS analysis data has been added as a figure panel in the Extended Data Figure 1j in the revised manuscript for your review.

14Extended Data Figure 1j

- 5) To study the oligomeric status of Rv3806c in its native membrane environment prior to the introduction of any detergent, we further performed structure-guided disulfide crossbridge experiment for the native *Msm* membrane-embedded

Rv3806c. D267 and T216 which mediate the protomer-protomer interface at the cytoplasmic side were selected for double cysteine mutation. Western blot of the cell membrane of *Msm* overexpressing the T216C/D267C mutant reveals that it primarily forms a trimer on the *Msm* membrane, in consistence with our EM and biochemical results. This has been described in the main-text of the revised MS under “Rv3806c is functional as a trimer” subtitle, and the data has been added as 3 figure panels in the Extended Data Figs. 1f-h.

Extended Data Figs. 1f-h

In Extended Figure 1e, why don't we observe isotopic distributions? The resolution of the spectrum in the figure is very poor, and it is not very convincing. Can authors do fragmentation to make a more convincing case? What does the blue arrow on the y-axis mean?

Response: Thanks for the professional comments.

Q1: In the original Extended Data Fig. 1e, the MS result was generated using peakviewer software under the Single Ion Monitor (SIM) mode without isotopic distributions. The blue arrow on the y-axis indicates the threshold level set in the software, only those with intensity values above the threshold were shown.

To address the isotopic distributions issue, we generated the mass spectrum result under the Full Scan mode, in such a mode, when zooming in around m/z of 777.6 (DP mass size), three isotopic distributions could be observed. The image quality has also been improved. The updated mass spectrometry result has been added as a figure panel in Extended Data Figure 1k in the revised manuscript for your review.

use, share original

where the authors are anonymous, such as is the case for the reports of anonymous peer reviewers, author attribution should be to 'Anonymous Referee' followed by a clear attribution to the source work. The images or other third party material in this file are included in the article's Creative Commons license, unless indicated otherwise in a credit line to the material. If material is not included in the article's Creative Commons license and your intended use is not permitted by statutory regulation or exceeds the permitted use, you will need to obtain permission directly from the copyright holder. To view a copy of this license, visit <http://creativecommons.org/licenses/by/4.0/>.

Extended Data Figure 1 kQ2: We did a fragmentation of the sample, with a DP standard as control. We attached in this response letter to show the fragmentation result to the Reviewer #2. Mass spectrum data shows the sample has the same fragmentation feature (PO_3^- , H_2PO_4^- , and DP) as the standard control, under the SIM mode and collision energy of 55 using an AB SCIEX TRIPLETOF 4600 System.

The higher turnover rate of K174T claimed in Extended Fig 9a is not convincing. Please provide a statistical analysis to substantiate this claim. In addition, we would like the authors to discuss an alternative possibility. It is equally possible that the periplasmic side of the enzyme is used for feedback inhibition by DPA. The mutations make the enzyme unresponsive to DPA, making it constitutively active. In such a scenario, you don't have to assume enhanced activities from the mutant enzymes. Instead, the authors should observe inhibition of the WT enzyme, but not mutant enzymes, by DPA in their enzyme assay. We don't think that the authors have to do this experiment, but the authors should discuss such an alternative possibility (which we think is actually more feasible and exciting).

Response: Firstly, we thank Reviewer#2 for pointing out the enzymatic data of K174T mutant as it appeared as similar level as the WT shown in Extended Data Fig. 9a. For the WT Rv3806c, the PRTase activity was measured as 11076.8 ± 1548.5 CPM from five technical replicates; and for the K174T mutant, the PRTase activity was measured as 12645.7 ± 1198.0 CPM from three technical replicates. Given that the increase in PRTase activity is minor, in the revised manuscript the K174T mutant were not specified.

	Exp 1	Exp 2	Exp 3	Exp 4	Exp 5
Rv3806c WT	10740	10856	13694	9568	10526

Rv3806c K174T	12035	11876	14026		
---------------	-------	-------	-------	--	--

Secondly, we very much appreciated the “DPA feedback inhibition” possibility suggested by Reviewer #2. We agreed with Reviewer #2 and thought this alternative possibility described by the reviewer is very exciting! Inspired by this, we further speculated that the POPG binding sites (also the mutation sites of EMB resistance) created by the protomer-protomer interface observed in the EM structure could be an allosteric site of the end-product DPA when the DPA level reaches high. We have

added this possibility into the discussion part of the revised MS. We believe that this may provide novel insights into the mycobacterial cell wall biosynthesis and regulatory mechanisms and requires further investigation of the field.

By looking at Fig. 5c, we are struck by the enzyme's inner core appearing to have a channel penetrating through the membrane all the way from cytoplasmic to periplasmic spaces. Is this true? If there is a channel in this enzyme, does a similar channel exist in other UbiA homologs? Is it possible that DP in the periplasmic side may also be acquired by this enzyme, possibly through this channel? If so, this enzyme can effectively function as a DP flippase, which we feel is an exciting possibility. Can authors clarify the structure in this regard and discuss it more in the manuscript?

Response: Yes indeed, the inner core forms a continuous channel almost throughout the membrane lipid bilayer, as represented in Fig.3a and Fig.5c. Structural analysis of other UbiA family enzymes including *Af. UbiA*, *Ap. UbiA* and human UBIAD1 reveals that none of them has a continuous TM channel like Rv3806c. The unique channel in Rv3806c is mainly due to the distinct TM arrangement of helical bundle 2 (TM6-9) of Rv3806c. Although we can only observe the DP density within the cytosolic side (Fig. 3c), the possibility that the periplasmic side of the channel serves as DP binding site cannot be ruled out. We therefore accepted this possibility that Rv3806c might also function as a DP flippase suggested by Reviewer #2. This possibility is added in the discussion part in the revised manuscript, as we currently have no more experimental data to further support this hypothesis. But we do think this is an interesting idea, which may appeal to a board readership and inspire new interests in the *Mtb* cell wall and cell membrane biology, as the DP flippase has not been identified yet.

We are not convinced with the proposed steps of the enzymatic reaction proposed in Fig. 5c. The authors proposes that DP comes in first based on the fact that the enzyme was purified with a bound DP. However, when they incubated the enzyme with PRPP, they no longer found DP in the enzyme structure. Their reasoning in Line 233-235 is reasonable, but that means that the enzyme can exist as a PRPP-bound form without DP. Based on these observations, it seems like the enzyme can accept either substrate first. To us, there appears to be no convincing evidence for the authors to suggest that DP binds to the enzyme before PRPP does.

Response: Thanks for pointing out this. We accepted Reviewer's comment. We have corrected our description in both main-text (under "Catalytic mechanism" paragraph) and related Fig. 5c in the revised manuscript to propose that either DP or PRPP can

bind in the active site prior to the other.Fig. 5c

Finally, there is no discussion on why this enzyme must be a trimer. Can the authors provide any insights into this particular finding? How might this arrangement aid in the mechanism of catalysis?

Response: Thanks for pointing out this. In this study we found Rv3806c protein samples in both GDN solution and nanodisc environment form a trimer. The authors could provide insights into this finding from two potential aspects. Firstly, the structure reveals that the helical bundle 2 of each protomer forms the core of the Rv3806c trimer, with the trimeric interface mainly mediated by TM6 and TM7, while the helical bundle 1 constitutes the peripheral part of the trimer. This assembly is proposed to favor a relatively rigid core region where the hydrophobic substrate DP is embedded and a relatively flexible peripheral region which undergoes conformational changes upon PRPP binding as observed in this study. Secondly, the phospholipid binding pocket which contributes to the protomer-protomer interface is on the opposite side of the substrate binding pockets, this could provide an allosteric site for lipid molecules (either phospholipids observed in the EM structure or the possible DPA binding site under the possibly negative feedback mechanism proposed by Reviewer #2) to regulate the reaction only upon forming an oligomer. This has been added into the discussion part in the revised manuscript.

Minor comments

Line 28: “Kingdom” is an outdated concept. “Domain” is more appropriate.

Response: thanks for the correction. We have corrected this in the revised manuscript.

Line 34: Why is it unexpected that PRPP is used in the biosynthesis of AG and LAM?
This was not clear to us.

Response: We described PRPP utilization in cytosolic metabolism of nucleotides, amino acids and cofactors in the first two sentences in the introduction part, which is “expected”. In mycobacteria, PRPP is also used in biosynthesis of AG and LAM, which is recognized by experts in *Mtb* cell wall field but not by all readership of Nature Microbiology, so “unexpectedly” is used here. We have reorganized the sentence here

by saying "PRPP is unexpectedly utilized in the biosynthesis of two key components of the bacterial cell wall" in the revised manuscript.

Line 68: We don't think GDN detergent is the key method information for the enzyme purification. The authors did not use GDN to purify Rv3806c. Rather, they used anti-FLAG resin to pull down FLAG-tagged Rv3806c. Perhaps, it is more appropriate to say that GDN was used to solubilize Rv3806c.

Response: Thanks for the correction. Yes, GDN is used as detergent in the purification buffer, anti-FLAG resin is used for affinity purification of Rv3806c. This has been corrected in the revised manuscript by saying "FLAG-tagged Rv3806c was expressed in M.smegmatis (Msm) and purified to homogeneity. Rv3806c in GDN solution was used for functional studies (Extended Data Figs. 1a-b)."

Line 69: Please provide a citation for your established cell-free PRTase assay.

Response: We apologize for the error here, "using our established ..." has been replaced by "using a previously established ...". And the reference is "Scherman MS, Kalbe-Bournonville L, Bush D, Xin Y, Deng L, McNeil M. Polyprenylphosphate-pentoses in mycobacteria are synthesized from 5-phosphoribose pyrophosphate. J Biol Chem. 1996 Nov 22;271(47):29652-8", which has been cited here.

Line 77: It is confusing to state that DP was clearly resolved because only a portion of DP is visible. The authors do clarify this point later in the manuscript. However, without knowing what the authors meant, this statement was confusing and misleading.

Response: Thanks for pointing out this. we apologized for making a confusing statement here. we have corrected this by saying "Also clearly resolved was the major part of the endogenous substrate DP which will be discussed later in details (Fig. 3c, Extended Data Fig. 4a, and Tables 1-2)."

Line 140: Fig. 3e – do you mean Fig. 3d? Note that the figure legend is also messed up.

Response: We apologized for the figure citation error here and the legend error for Fig.3. We have corrected the Figure citation to "Fig. 3d" in the revised manuscript. The original legends for Fig. 3e and 3f have been corrected as Fig. 3d and 3e in the revised manuscript.

The figure legend for Fig. 3c has been corrected as: "*(c) The structure of (left) DP-bound Rv3806c, and (right) structures of PP-prenyl bound AfUbiA and ApUbiA, proteins are shown in ribbon representation and ligands are shown as spheres. The protein color is the same as 3b. The ligands are shown as spheres and colored by atom: oxygen in red, phosphate in orange, and carbon of DP in gold.*"

Line 143: There is no Fig. 3f.

Response: Thanks, this should be *Fig. 3e*, we have addressed this issue in the previous comment and corrected the figure citation in the revised manuscript.

Line 149: It should be Fig. 3b. Review your manuscript more carefully before submission.

Response: Here we discussed the difference between the PP-prenyl binding pocket of UbiA superfamily enzymes and the DP binding pocket of Rv3806c, which is shown in Fig. 3c from a side-view within the membrane, rather than Fig. 3b shown from a top view.

Line 166: Please indicate Extended Figure 6a to support this sentence.

Response: Thanks, we have cited Extended Data Fig. 6a to support the MST result here in the revised manuscript.

Line 170: Please indicate Extended Figure 6b specifically.

Response: Thanks, Extended Data Fig. 6b has been specified here.

Line 191: Did you mean to reference Figure 4b? If you really want us to look at Fig. 4c, please indicate Lys28, Tyr70, and Tyr157 in this panel. Also, 3-letter code for tyrosine is “Tyr” and not “Try”.

Response: We apologized for the figure citation error and type error here. Yes for the interaction with 5'-phosphate group of PRPP, we wanted to indicate Fig. 4b, and Tyr70 and Tyr157 type errors have been corrected in the revised manuscript.

Line 212: POPG was never defined anywhere. Please define.

Response: Thanks for pointing out this, we have added the full name *palmitoyl-oleoyl-phosphatidylglycerol* here as it's the first time it appears in the manuscript. We also provided the source of POPG in the “Protein purification and nanodisc reconstitution” paragraph in the methods.

Line 316: It is inaccurate to state that cell wall biosynthesis in Mtb (as a whole) is catalyzed by Rv3806c.

Response: Thanks. We accepted this comment. The words “The latter is catalyzed by” have been deleted from legend for figure panel 1a. Legend for figure panel 1b has been revised as “ (b) Rv3806c is involved in the latter by catalyzing phosphoribose transfer from PRPP to a membrane-anchored substrate DP to generate DPPR, an essential precursor of arabinosyl donor DPA for AG and LAM biosynthesis. ”

Line 346: “Normalized PRTase analysis” is not descriptive enough. This line should say “Normalized PRTase activity analysis” or something similar.

Response: Thanks for the correction. To clearly show the PRTase activity of WT and all mutant Rv3806c proteins in this study, we have represented all PRTase activity data as raw data, using the unit “count per minute (CPM)” in all figure panels regarding to PRTase activity. There is no normalized data any more. In the revised manuscript, the “Normalized” word has been deleted here and all PRTase data hereafter.

Line 353, 354: Can you specify the type of molecular interactions that are indicated by the dashed lines?

Response: Yes, we have updated 4b and 4c figure panels and related legends, using different lines to show different types of interactions in the revised manuscript. Specifically, in Fig. 4b, yellow dashed lines indicate hydrogen bond with the phosphoribose, dark blue dashed lines indicate Van der Waals' force with the phosphoribose based on their distances. In fig. 4c, black lines indicate the Mg²⁺ coordination involving sidechains and oxygen atoms of the pyrophosphate group. Yellow dashed lines indicate hydrogen bond network with the pyrophosphate group. The figures 4b and 4c have been updated for your review.

Fig. 4b

Fig. 4c

Line 354: The bound substrate in the structure contains a PPi moiety, but it is inaccurate to refer to this substrate as PPi as if it were a free pyrophosphate.

Response: Thanks a lot for the correction. We have changed PPi to “*pyrophosphate group*” here and hereafter in the revised manuscript when discussing the moiety within the PRPP substrate.

Figure 1a. While studies have shown LAM to be surface exposed, LAM's size makes it unlikely to span from the inner membrane through the outer membrane (in fact, the position of the arabinan domain of LAM intercalated with outer membrane lipids as shown here is not supported in the literature). Additionally, AG should be shown to be covalently linked to the outer membrane in this cartoon. It is also a bizarre way of color-coding, like blue “M” referring to mannan. Finally, the plasma membrane is misspelled as “plasm membrane”.

Response: We appreciated Reviewer #2 for this very professional comment. We have revised Figure 1a accordingly for your review.

Fig. 1a

Figure 2f. Why are there four dots from WT if it was done in triplicate? The bar graph presumably shows the average, but it does not look like the average of the four dots. What are the data normalized to?

Response: Thanks and we wanted to clarify that the PRTase activity data for the WT Rv3806c in this manuscript (including Figs. 1c (the 2nd lane), 2f, 3e, 4d, and Extended Data Fig. 9a) are the same data: the average from 5 technically independent assays using purified WT Rv3806c. We have regenerated the PRTase activity data represented by raw data, using the unit CPM, as previously demonstrated in our response to reviewer's comment on Line 346. We apologized for the error in data statistics and we have carefully addressed this issue in all related figure panels and legends in the revised manuscript.

Decision Letter, first revision:

Message 20th October 2023

29:
Dear Professor Rao

Thank you for submitting the revised version of your Article entitled "Structure and mechanism of a membrane-bound phosphoribosyltransferase central to M. tuberculosis cell wall biosynthesis" for consideration in Nature Microbiology. After careful consideration and discussion with my editorial colleagues, we have decided that we will not be sending the manuscript back to our referees in its present form.

While we remain very interested in this work, and do appreciate the revisions made and additional work included in the latest version of the manuscript, there is one outstanding issue which we need clarification on. One of the referees had previously raised some concerns about the PRTase activity data for the WT Rv3806c. We appreciate that you have sent in the data in an excel file, but would urge you to provide images of the source autoradiographs that provided this data. In additional discussions with the referee they had emphasised the need for WT enzymatic activity to be tested alongside the mutants to which it is being compared in each dataset. We would like you to verify that this is the case before we make a decision on whether or not to send this manuscript back to the reviewers.

Nature Microbiology is committed to improving transparency in authorship. As part of our efforts in this direction, we are now requesting that all authors identified as 'corresponding author' on published papers create and link their Open Researcher and Contributor Identifier (ORCID) with their account on the Manuscript Tracking System (MTS), prior to acceptance. This applies to primary research papers only. ORCID helps the scientific community achieve unambiguous attribution of all scholarly contributions. You can create and link your ORCID from the home page of the MTS by clicking on 'Modify my Springer Nature account'. For more information please visit www.springernature.com/orcid.

I am sorry that we cannot respond more positively on this occasion but do hope that you are able to address the points noted above and submit a revised version of your manuscript for further consideration.

Author Rebuttal, first revision:

Reviewer Expertise:

30Referee #1: CryoET

Referee #2: Mycobacterial cell wall and membrane biology

Reviewer Comments:

Reviewer #1 (Remarks to the Author):

The paper describes the structure of the membrane-bound Rv3806c (spans the cytoplasm and periplasm), a phosphoribosyl transferase, whereby ligand PRPP targets membrane bound ligand decaprenol phosphate (DP), to generate the product ligand DPPR and release of pyrophosphate. Rv3806c is composed of 9 transmembrane (TM) helices divided into two helical bundles that are connected through periplasmic and cytoplasmic loops. Notably, Rv3806c adopts a trimeric alignment stabilized by protomer-protomer interfaces and lipids, with the trimeric state critical in the mediation of phosphoribose transfer. The authors solved the structure of Rv3806c bound to DP and could observe the phosphate and some of the prenyl linkage positioned in a cavity within the TM segment connected to an extramembrane cleft opening to the cytoplasm. The DP-binding cavity in Rv3806c is distinct from its counterpart in UbiA superfamily enzymes. They also solved the structure of Rv3806c bound to endogenous PRPP-Mg ligand, wherein the phosphate group is buried in the membrane. Key residues contacting both bound DP and bound PRPP were mutagenized to monitor their impact on catalysis. Based on the positioning of DP and PRPP, an inverted catalytic mechanism was proposed for formation of product DPPR.

Overall, this was a challenging structural project brought to successful conclusion, which was complemented by mutagenesis studies to provide insights into the catalytic mechanism. I recommend publication in its current form but would encourage the authors in future studies to try and solve the structure of Rv3806c bound to product DPPR, towards a fuller understanding of the catalytic cycle.

Response: We very much appreciated Reviewer #1 for the supportive comments to our manuscript and suggestions on future studies of Rv3806c towards a fuller understanding of the catalytic cycle. The difficulty in solving the structure of Rv3806c with its product DPPR is the challenging chemosynthesis of DPPR which contains a highly hydrophobic decaprenyl tail. We aim to solve the DPPR-bound structure and look forward to the success on chemosynthesis of DPPR .

31Reviewer #2 (Remarks to the Author):

This study reports the cryo EM structure of Rv3806c, one of the critical enzymes for the synthesis of arabinose donor in mycobacteria. The detailed structural and biochemical analysis provides exciting new insights on how this unusual membrane-embedded enzyme utilizes PRPP to produce DPPR. In particular, cryo-EM structures of the enzyme with bound substrates provide interesting mechanistic understanding of this very important enzyme. Another notable achievement is the establishment of the enzymatic assay system for this membrane-embedded enzyme. Overall, the study was rigorously done, and the results are generally supportive of the author's conclusions. However, we have several major questions and thoughts that the authors can clarify further.

Response: We appreciate Reviewer #2 for his/her overall supportive comments to our work. By reading the following comments, the authors recognized Reviewer #2 as a real expert in *Mtb* cell wall biology, and the suggestions will greatly help improve the manuscript quality to merit publication in Nature Microbiology.

Major comments

We are not convinced of the quality of the data presented in Fig. 2e. The native SDS-PAGE gel is without standards, and it is not even clear if these two smeary bands are different in mobility. Can the authors run both L231W and WT together in a single lane on the native PAGE to show that they can be separated in a single lane? SEC is supposed to be more quantitative than native PAGE, but the authors did not show the molecular weight markers. The authors claim that T227W came out in the void volume, but they did not provide any evidence for this. There is also no evidence for them to claim that T227W was denatured. The suggestion that L231W is not forming a complex equivalent to the WT trimer is acknowledged based on the SEC data. However, the data is not sufficient to claim that L231W is properly folded. Can the authors do cryo-EM of this mutant, or at least CD or some other structural analysis to show that it's not denatured? Overall, Fig. 2e is of poor quality, and needs to be revised.

Response: We thank Reviewer 2 for pointing out the trimeric assembly issue. Based on his/her suggestion, we have performed the following experiments:

- 1) We have improved the quality of the native PAGE to better show the difference between the WT-Rv3806c band and the L231W mutant band, together with molecular weight standards. To be noted, both the WT-Rv3806c and the mutant bands showed much higher molecular weights in the native PAGE than the theoretical molecular weights (96kD for trimer, 64kD for dimer and 32kD for monomer). This is due to the fact that membrane proteins purified in detergent micelles (GDN in this study) are commonly larger than their theoretical molecular

weight. The updated BN PAGE data was added as a new panel in Extended Data Fig. 1h (together with the newly added dimer interface cross-linking data to study the oligomeric state of Rv3806c in its native membrane) in the revised manuscript. Given that the WT-Rv3806c and the L231W mutant are well separated in the native PAGE, the two samples were not mixed in the single lane.

Extended Data Fig. 1h

- 2) We have now numerically labelled the elution volume of WT-Rv3806c and mutants samples in the SEC data, so it can be now easily observed that T227W (9.02 ml) eluted close to the void volume in the same purification buffer as the WT-Rv3806c (10.06 ml) sample. This result has now been updated in Figure panel 2e in the revised manuscript for your review.

Fig. 2e

- 3) We have added the negative staining EM analysis of WT-Rv3806c and mutant Rv3806c samples purified in GDN solution to study their oligomeric states in the revised manuscript. The representative micrograph of negative stained WT-Rv3806c shows homogeneous particles. Reference-free 2D class averages of particles revealed a 3-fold symmetric feature. The data indicated that WT-Rv3806c forms a trimer in GDN solution, consistent with the cryo-EM structure of the nanodisc reconstituted WT-Rv3806c sample. The representative micrograph of the negative stained L231W mutant showed smaller particles, as well as smaller 2D class averages of particles compared to the WT-Rv3806c sample; it is technically difficult to do further cryo-EM data analysis. To be noted, at this resolution it is difficult to distinguish whether the L231W mutant sample is monomer or dimer,

so we concluded that this mutant forms a lower oligomer than the trimeric WT-Rv3806c sample. The representative micrograph of negative stained T227W mutant showing particles that appear as aggregates, consistent with its behavior in the SEC analysis, whereby it was eluted close to the void volume. Therefore, in the revised manuscript, we have concluded that the T227W mutant was improperly folded. The negative staining EM analysis of WT-Rv3806c, L231W and L227W mutants have been added as a figure panel in the Extended Data Figure 1i in the revised manuscript.

Extended Data Figure 1i

- 4) We have added microfluidic modulation spectroscopy (MMS) analysis to our study of the secondary structural features of the L231W mutant. The MMS data shows that both WT-Rv3806c and the L231W mutant are properly folded in GDN solution with an alpha-helix dominated structure. The minor difference between the WT-Rv3806c and the L231W mutant may be due to the detramerization. The MMS analysis data has been added as a new figure panel in the Extended

Open Access
This article is licensed under a Creative Commons Attribution 4.0 International License, which permits use, sharing, adaptation, distribution and reproduction in any medium or format, as long as you give appropriate credit to the original author(s) and the source, provide a link to the Creative Commons licence, and indicate if changes were made. The images or other third party material in this article are included in the article's Creative Commons licence, unless indicated otherwise in a credit line to the material. If material is not included in the article's Creative Commons licence and your intended use is not permitted by statutory regulation or exceeds the permitted use, you will need to obtain permission directly from the copyright holder. To view a copy of this licence, visit <http://creativecommons.org/licenses/by/4.0/>.

license, which permits appropriate credit to the original author(s) and the source, provide a link to the Creative Commons licence, and indicate if changes were made. The images or other third party material in this article are included in the article's Creative Commons licence, unless indicated otherwise in a credit line to the material. If material is not included in the article's Creative Commons licence and your intended use is not permitted by statutory regulation or exceeds the permitted use, you will need to obtain permission directly from the copyright holder. To view a copy of this licence, visit <http://creativecommons.org/licenses/by/4.0/>.

Data Figure 1j in the revised manuscript.

Extended Data Figure 1jLegend: **j**, MMS data showed both WT-Rv3806c and L231W were folded in solution with an alpha-helix dominated structure. The absolute spectra (top left) and second derivative spectra (bottom left) show the main peaks around $1657\text{-}1658\text{ cm}^{-1}$ which is a signature peak for alpha-helix structure in the Amide I band. The percentage secondary structural elements were calculated by Gaussian curve fitting on the baseline-corrected second derivative spectra using the peak assignments designated in top right. (Bottom right) Higher Order Structure (HOS) fractional contributions of the L231W mutant and WT-Rv3806c protein.

- 5) To study the oligomeric status of WT-Rv3806c in its native membrane environment prior to the introduction of detergent, we have further performed structure-guided disulfide cross-bridge experiments for the native *Msm* membrane-embedded Rv3806c. D267 and T216 which mediate the protomer- protomer interface at the cytoplasmic side were selected for double cysteine mutation. Western blot of the cell membrane of *Msm* overexpressing the T216C/D267C mutant reveals that it primarily forms a trimer on the *Msm* membrane, consistency with our EM and biochemical results. This is now described in the main-text of the revised manuscript under “Rv3806c is functional as a trimer” subtitle, and the data has been added as 3 new figure panels in the Extended Data Figs. 1f-h.

Extended Data Figs. 1f-h

Legend: **f**, D267 and T216 which mediate the protomer-protomer interface at the cytoplasmic side were selected for double cysteine crossbridge. **g**, Western blot analysis of cross-linking in the absence and presence of detergent, using anti-FLAG antibody.

Lane 1-2, *Msm* cell membranes of T216C-D267C mutant; lane 3-4, T216C-D267C mutant purified in GDN; lane 5-6, *Msm* cell membranes of WT-Rv3806c as control; lane 7: molecular weight marker. All samples were analyzed with or without 10mM reducing agent DTT. The signal of the dimer band in lane 1,2 and 4 may be due to insufficient disulfide bond formation between only two protomers. **h**, Blue native PAGE analysis of the WT-Rv3806c and mutant Rv3806c samples purified in GDN, the T216C-D267C mutant appears as a trimer as same as the WT-Rv3806c, whereas the molecular weight of the L231W mutant is lower than the WT-Rv3806c.

In Extended Figure 1e, why don't we observe isotopic distributions? The resolution of the spectrum in the figure is very poor, and it is not very convincing. Can authors do fragmentation to make a more convincing case? What does the blue arrow on the y-axis mean?

Response: We thank the Reviewer for his/her comments.

Q1: In the original Extended Data Fig. 1e, the MS data was generated using peak viewer software under Single Ion Monitor (SIM) mode without isotopic distribution. The blue arrow on the y-axis indicates the threshold level set in the software, only those with intensity values above the threshold were shown. To address the isotopic distribution issue, we have generated the mass spectrum data under the Full Scan mode, and in such a mode, when zooming in around m/z of 777.6 (DP mass size), three isotopic distributions can be observed. The image quality has now also been improved. The updated mass spectrometry data has been added as a new figure panel in Extended Data Figure 1k in the revised manuscript.

Extended Data Figure 1 k

Legend: **k**, (left) Full scan mode mass spectrometry analysis of solvent extracted DP from Rv3806c purified in GND solution, the ion of m/z 777.6, which corresponds to the peak of DP, is indicated, (middle) isotopic distributions of DP are indicated upon m/z ranging from 750 to 800 Da. (right) MS intensity of DP was measured in a dose dependent manner to protein concentration.

Q2: We have performed fragmentation of the sample, with a DP standard as a control.

40We have attached in this response letter to show the fragmentation result to Reviewer #2. Mass spectrum data shows the sample has the same fragmentation features (PO_3^- , H_2PO_4^- , and DP) as the standard control, under the SIM mode and collision energy of 55 using an AB SCIEX TRIPLETOF 4600 System.The higher turnover rate of K174T claimed in Extended Fig 9a is not convincing. Please provide a statistical analysis to substantiate this claim. In addition, we would like the authors to discuss an alternative possibility. It is equally possible that the periplasmic side of the enzyme is used for feedback inhibition by DPA. The mutations make the enzyme unresponsive to DPA, making it constitutively active. In such a scenario, you don't have to assume enhanced activities from the mutant enzymes. Instead, the authors should observe inhibition of the WT enzyme, but not mutant enzymes, by DPA in their enzyme assay. We don't think that the authors have to do this experiment, but the authors should discuss such an alternative possibility (which we think is actually more feasible and exciting).

Response: Firstly, we thank Reviewer#2 for pointing out the enzymatic data of the K174T mutant as it appeared at a similar level as WT-Rv3806c as shown in Extended Data Fig. 9a. For the WT-Rv3806c, the PRTase activity was measured as 11575.5 ± 1588.9 CPM from two independent replicates; and for the K174T mutant, the PRTase activity was measured as 12645.7 ± 1198.0 CPM from three technical replicates. Given that the increase in PRTase activity is negligible, in the revised manuscript the K174T mutant was not extensively discussed.

	Exp 1	Exp 2	Exp 3
Rv3806c WT	10452	12699	
Rv3806c K174T	12035	11876	14026

Secondly, we very much appreciated the “DPA feedback inhibition” possibility as suggested by Reviewer #2. We agree with Reviewer #2 and think that this alternative

42possibility is very exciting! Inspired by this, we have further speculated that the POPG binding sites (also the mutation sites of EMB resistance) created by the protomer-protomer interface observed in the EM structure could be an allosteric site of the end-product DPA, when DPA reach high levels. We have now added this possibility into the *discussion part* of the revised MS. We believe that this may provide novel insights into mycobacterial cell wall biosynthesis and regulatory mechanisms and requires further investigation in the field.By looking at Fig. 5c, we are struck by the enzyme's inner core appearing to have a channel penetrating through the membrane all the way from cytoplasmic to periplasmic spaces. Is this true? If there is a channel in this enzyme, does a similar channel exist in other UbiA homologs? Is it possible that DP in the periplasmic side may also be acquired by this enzyme, possibly through this channel? If so, this enzyme can effectively function as a DP flippase, which we feel is an exciting possibility. Can authors clarify the structure in this regard and discuss it more in the manuscript?

Response: Yes indeed, the inner core forms a continuous channel, almost throughout the membrane lipid bilayer, as represented in Fig. 5c. Structural analysis of other UbiA family members, including *Af. UbiA*, *Ap. UbiA* and human UBIAD1, reveals that none have a continuous TM channel, like Rv3806c. The unique channel in Rv3806c is mainly due to the distinct TM arrangement of helical bundle 2 (TM 6-9) of Rv3806c. Although, we can only observe the DP density within the cytosolic side (Fig. 3c), the possibility that the periplasmic side of the channel serves as a DP binding site cannot be ruled out. We therefore accept the possibility that Rv3806c might also function as a DP flippase as suggested by Reviewer #2. This possibility has now been added in the discussion part in the revised manuscript, as we currently have no more experimental data to further support this hypothesis. But we do think that this is an interesting hypothesis, which may appeal to a boarder readership and inspire new interests in the *Mtb* cell wall and cell membrane biology, as the DP flippase has not been identified to date.

We are not convinced with the proposed steps of the enzymatic reaction proposed in Fig. 5c. The authors proposes that DP comes in first based on the fact that the enzyme was purified with a bound DP. However, when they incubated the enzyme with PRPP, they no longer found DP in the enzyme structure. Their reasoning in Line 233-235 is reasonable, but that means that the enzyme can exist as a PRPP-bound form without DP. Based on these observations, it seems like the enzyme can accept either substrate first. To us, there appears to be no convincing evidence for the authors to suggest that DP binds to the enzyme before PRPP does.

Response: We thank the Reviewer for pointing this out and we accept his/her comments. We have modified our description in both the main-text (under the "Catalytic mechanism" paragraph) and related Fig. 5c in the revised manuscript to propose that either DP or PRPP can bind in the active site prior to the other.

Fig. 5c Proposed catalytic model of Rv3806c

Finally, there is no discussion on why this enzyme must be a trimer. Can the authors provide any insights into this particular finding? How might this arrangement aid in the mechanism of catalysis?

Response: We thank the Reviewer for his/her comments. In this study we have found Rv3806c protein samples in both the GDN solution and nanodisc environment form a trimer. In terms of further insights, firstly, the structure reveals that the helical bundle 2 of each protomer forms the core of the Rv3806c trimer, with the trimeric interface mainly mediated by TM6 and TM7, while the helical bundle 1 constitutes the peripheral part of the trimer. This assembly is proposed to favor a relatively rigid core region where the hydrophobic substrate DP is embedded and a relatively flexible peripheral region which undergoes conformational changes upon PRPP binding as observed in this study. Secondly, the phospholipid binding pocket which contributes to the protomer-protomer interface is on the opposite side of the substrate binding pockets, this could provide an allosteric site for lipid molecules (either phospholipids observed in the EM structure or the possible DPA binding site under the possibly negative feedback mechanism proposed by Reviewer #2) to regulate the reaction only upon forming an oligomer. This has now been added into the *discussion part* in the revised manuscript.

Minor comments

Line 28: “Kingdom” is an outdated concept. “Domain” is more appropriate.

Response: We have corrected this in the revised manuscript.

Line 34: Why is it unexpected that PRPP is used in the biosynthesis of AG and LAM?
This was not clear to us.

Response: We described PRPP utilization in cytosolic metabolism of nucleotides, amino acids and cofactors in the first two sentences in the introduction, which is

“expected”. In mycobacteria, PRPP is also used in biosynthesis of AG and LAM, which is recognized by experts in *Mtb* cell wall field but will not be familiar to all the readership of Nature Microbiology, so “unexpectedly” is used here. We have reorganized the sentence here by saying “*PRPP is unexpectedly utilized in the biosynthesis of two key components of the bacterial cell wall*” in the revised manuscript.

Line 68: We don’t think GDN detergent is the key method information for the enzyme purification. The authors did not use GDN to purify Rv3806c. Rather, they used anti-FLAG resin to pull down FLAG-tagged Rv3806c. Perhaps, it is more appropriate to say that GDN was used to solubilize Rv3806c.

Response: We thank the Reviewer for this correction. Yes, GDN was used as detergent in the purification buffer, the anti-FLAG resin was used for affinity purification of Rv3806c. This has been corrected in the revised manuscript by saying “*FLAG-tagged Rv3806c was expressed in M.smegmatis (Msm) and purified to homogeneity. Rv3806c in GDN solution was used for functional studies (Extended Data Figs. 1a-b).*”

Line 69: Please provide a citation for your established cell-free PRTase assay.

Response: We apologize for the error here, “using our established ...” has been replaced by “using a previously established ...”. and the reference is “Scherman MS, Kalbe-Bournonville L, Bush D, Xin Y, Deng L, McNeil M. Polyprenylphosphate-pentoses in mycobacteria are synthesized from 5-phosphoribose pyrophosphate. J Biol Chem. 1996 Nov 22;271(47):29652-8”, which has been cited here.

Line 77: It is confusing to state that DP was clearly resolved because only a portion of DP is visible. The authors do clarify this point later in the manuscript. However, without knowing what the authors meant, this statement was confusing and misleading.

Response: We thank the Reviewer for pointing out this and apologize for the confusing statement. We have corrected this by saying “*Also clearly resolved was the major part of the endogenous substrate DP which will be discussed later in details (Fig. 3c, Extended Data Fig. 4a, and Tables 1-2).*”

Line 140: Fig. 3e – do you mean Fig. 3d? Note that the figure legend is also messed up.

Response: We apologize for the figure citation error here and the legend error for Fig. 3. We have corrected the Figure citation to “*Fig. 3d*” in the revised manuscript. The original legends for Fig. 3e and 3f have been corrected as Fig. 3d and 3e in the revised manuscript.

The figure legend for Fig. 3c has been corrected as: “*(c) The structure of (left) DP-bound Rv3806c, and (right) structures of PP-prenyl bound AfUbiA and ApUbiA, proteins are shown in ribbon representation and ligands are shown as spheres. The*”

protein color is the same as 3b. The ligands are shown as spheres and colored by atom: oxygen in red, phosphate in orange, and carbon of DP in gold.

Line 143: There is no Fig. 3f.

Response: This should be *Fig. 3e*, we have addressed this issue in the previous comment and corrected the figure citation in the revised manuscript.

Line 149: It should be Fig. 3b. Review your manuscript more carefully before submission.

Response: Here we discussed the difference between the PP-prenyl binding pocket of UbiA superfamily enzymes and the DP binding pocket of Rv3806c, which is shown in Fig. 3c from a side-view within the membrane, rather than Fig. 3b shown from a top view.

Line 166: Please indicate Extended Figure 6a to support this sentence.

Response: We have cited *Extended Data Fig. 6a* to support the MST result here in the revised manuscript.

Line 170: Please indicate Extended Figure 6b specifically.

Response: We specifically refer to Extended Data Fig. *6b* in the revised manuscript.

Line 191: Did you mean to reference Figure 4b? If you really want us to look at Fig. 4c, please indicate Lys28, Tyr70, and Tyr157 in this panel. Also, 3-letter code for tyrosine is “Tyr” and not “Try”.

Response: We apologize for the figure citation and type errors. Yes, for the interaction with 5'-phosphate group of PRPP, we wanted to indicate Fig. 4b, and the Tyr70 and Tyr157 type errors have been corrected in the revised manuscript.

Line 212: POPG was never defined anywhere. Please define.

Response: We thank the Reviewer for pointing this out; we have added the full name *palmitoyl-oleoyl-phosphatidylglycerol* here as it's the first time it appears in the manuscript. We have also provided the source of POPG in the “Protein purification and nanodisc reconstitution” paragraph in the methods.

Line 316: It is inaccurate to state that cell wall biosynthesis in Mtb (as a whole) is catalyzed by Rv3806c.Response: We accept this comment. The words “The latter is catalyzed by” have been deleted from the legend for figure panel 1a. Legend for figure panel 1b has been revised as: “ (b) Rv3806c is involved in the latter by catalyzing phosphoribose transfer from PRPP to a membrane-anchored substrate DP to generate DPPR, an essential precursor of arabinosyl donor DPA for AG and LAM biosynthesis. ”

Line 346: “Normalized PRTase analysis” is not descriptive enough. This line should say “Normalized PRTase activity analysis” or something similar.

Response: To clearly show the PRTase activity of WT-Rv3806c and all mutant proteins in this study, we have represented all PRTase activity data now as raw data, using the unit “count per minute (CPM)” in all figure panels regarding PRTase activity. There is no normalized data any more. In the revised manuscript, the “Normalized” word has been deleted here and all PRTase data hereafter.

Line 353, 354: Can you specify the type of molecular interactions that are indicated by the dashed lines?

Response: Yes, we have updated Fig. 4b and 4c figure panels and related legends, using different lines to show different types of interactions in the revised manuscript. Specifically, in Fig. 4b, yellow dashed lines indicate hydrogen bond with the phosphoribose, dark blue dashed lines indicate Van der Waals’ force with the phosphoribose based on their distances. In fig. 4c, black lines indicate the Mg²⁺ coordination involving sidechains and oxygen atoms of the pyrophosphate group. Yellow dashed lines indicate hydrogen bond network with the pyrophosphate group. The figures 4b and 4c have been updated for your review.

Fig. 4b

Fig. 4c

Legend: (b) Structure of PRPP-bound Rv3806c, protein is shown in ribbon representation, sidechains of phosphoribose binding sites and PRPP are shown as sticks, Mg^{2+} is shown as spheres. Yellow dashed lines indicate hydrogen bond with the phosphoribose, dark blue dashed lines indicate Van der Waals' force with the phosphoribose based on their distances. (c) Binding sites of Mg^{2+} and the pyrophosphate group. Black lines indicate the Mg^{2+} coordination involving sidechains and oxygen atoms of the pyrophosphate group. Yellow dashed lines indicate hydrogen bond network with the pyrophosphate group, sidechains and PRPP are shown as sticks, Mg^{2+} is shown as spheres.

Line 354: The bound substrate in the structure contains a PPI moiety, but it is inaccurate to refer to this substrate as PPI as if it were a free pyrophosphate.

Response: We have changed PPI to “*pyrophosphate group*” here and hereafter in the revised manuscript when discussing the moiety within the PRPP substrate.

Figure 1a. While studies have shown LAM to be surface exposed, LAM’s size makes it unlikely to span from the inner membrane through the outer membrane (in fact, the position of the arabinan domain of LAM intercalated with outer membrane lipids as shown here is not supported in the literature). Additionally, AG should be shown to be covalently linked to the outer membrane in this cartoon. It is also a bizarre way of color-coding, like blue “M” referring to mannan. Finally, the plasma membrane is misspelled as “plasm membrane”.

Response: We appreciated Reviewer #2 for this his/her comments and we have revised **Figure 1a** accordingly.

Fig. 1a

Figure 2f. Why are there four dots from WT if it was done in triplicate? The bar graph presumably shows the average, but it does not look like the average of the four dots. What are the data normalized to?

Response: We would firstly like to apologize to the Reviewer in terms of the PRTase data-sets for each experiment for not being clear. We have now provided an Extended Data XLS File Tables 1 to 5, which clearly shows the number of experiments *per* panel

as reported in the revised manuscript for the WT-Rv3806c and mutant samples. In addition, as mentioned above we have now removed all normalized data and have reported the raw data in all panels which refer to PRTase activity.

In addition, we would also like to apologize to the Reviewer, as we have just noted that the text in the materials section of the manuscript would imply that we ran TLC-autoradiograms for every PRTase assay sample; this was an oversight on our part, and we would like to firstly clarify this, and secondly provide a revised materials section which is more precise in terms of the PRTase experiments and reassure the Reviewer as follows:

In our PRTase assays, we firstly established that our control PRTase assay (minus WT-Rv3806c plus exogenous decaprenol phosphate) produced no detectable product by TLC (which is also confirmed by the very low radioactive background counts of 37 cpm), whereas our WT-Rv3806c PRTase assay (i.e. WT-Rv3806c sample plus exogenous decaprenol phosphate, recorded 10,740 cpm) from Experiment 1 from the Extended Data XLS File Table 1 produced only one product, DPP[¹⁴C]R, which was resolved from the [¹⁴C]ribose phosphate at the origin of the TLC, which was generated *in situ* from the DPP[¹⁴C]R enzymatic product due to the TLC-chromatography procedure. In our view, based on the clean and striking outcome of the data in Extended Data Fig. 1c in terms of minus/plus Rv3806c and the singular DPP[¹⁴C]R product, we then processed each time all subsequent PRTase assays with the Rv3806c mutants and additional WT-Rv3806c proteins directly to scintillation counting without the need for TLC-autoradiography; therefore, we performed scintillation counting on the entire assay sample in each case. Hence, we have now provided the XLS file with the raw cpm data for each sample/experiment and clearly show which two initial samples were processed for TLC-autoradiography. We feel this confusion stems from our text in the materials section, which should have been more precise in terms of which samples were processed for the TLC-autoradiography and which were completely counted as discussed above. I hope this clarifies why we have not provided TLC-autoradiograms for each sample, and we have now amended the text in the materials section in the revised manuscript to reflect this more accurately as follows:

“P[¹⁴C]RPP was prepared enzymatically from [¹⁴C]glucose as described⁶⁶. Each assay was performed in a final volume of 80 μL containing 50 mM MOPS buffer (pH 7.9),

P[¹⁴C]RPP (50,000 cpm), 80 μM DP, 2 μM Rv3806c, and 0.08% CHAPS. In some assays, divalent cations (2 mM) and ethambutol (100 μM) were added. Samples were incubated at 37°C for 1 hour, quenched by the addition of 6 mL of chloroform/methanol/water (10:10:3, v/v/v), thoroughly mixed, and 2.65 mL of chloroform and 1.125 mL of water added to each sample. The resulting bi-phasic mixture was vortexed, centrifuged and the lower organic layer recovered, washed thrice using 3 mL of chloroform/methanol/water (3:47:48, v/v/v) and dried. The PRTase control assay(minus WT Rv3806c + decaprenol phosphate) and the PRTase assay (WTRv3806c + decaprenol phosphate) were dried (Extended Data Fig. 1c and Extended Data XLS Table 1, PRTase assays from Experiment 1) and the reaction products resuspended in 100 μ L of chloroform/methanol/water (10:10:3, v/v/v) and an aliquot removed for scintillation counting to determine the incorporation of P[14 C]RPP into the radiolabeled reaction product DPP[14 C]R. A further aliquot from these two samples were analyzed using aluminum backed silica gel 60 F₂₅₄ thin layer chromatography (TLC) plates developed in chloroform/methanol/water/ammonium hydroxide (65:25:3.6:0.5, v/v/v/v) and visualized by autoradiography using Kodak BioMAX MR films (Extended Data Fig. 1c and Extended Data XLS Table 1, PRTase assays from Experiment 1). Since, the control lane (minus WT Rv3806c + decaprenol phosphate) produced no detectable product, whereas the WT-Rv3806c (+ decaprenol phosphate) sample produced only one product, DPP[14 C]R by TLC-autoradiography as shown in Extended Data Fig. 1c, the entire assay product for all subsequent PRTase assays for each of the Rv3806c mutants (Extended Data XLS File Tables 2-5, Experiments 1, 2 and 3) and additional WT-Rv3806c protein samples (Extended Data XLS File Tables 1-4, Experiments 1, 2 and 3; and Extended Data XLS File Table 5, Experiments 1 and 2) were subjected directly to scintillation counting without the need for TLC-autoradiography. As per the Extended Data XLS Tables 1-5, each assay was repeated in triplicate, except for Extended Data Fig. 9a, where the WT-Rv3806c PRTase activity was performed in duplicate.

Furthermore, we can confirm that the WT-Rv3806c enzymatic activity was tested every time alongside each of the mutants when the PRTase assay was performed, as shown in the Extended Date XLS File Tables 2, 3, 4 and 5 in the revised manuscript as follows:

- a. Table 2 for Fig. 2f, the WT-Rv3806c PRTase assays were performed alongside the mutant L231W, and other mutants were performed at the same time as L231W and are reported in Fig.3e and Fig. 4d. This is clearly mentioned in the footnotes of each Table.
- b. Table 3 for Fig. 3e, the WT-Rv3806c PRTase assays were performed alongside the mutants K28A, K191A and Y157A, and others reported in Fig. 2f and Fig. 4d, this is again mentioned for clarity in the footnote of the Table.
- c. Table 4 for Fig. 4d, the WT-Rv3806c PRTase assays were performed alongside the mutants D77, N73A, Q135A, K28A, Y70A, Y70F, Y157A, K143A, Y138A, Y138F, R22A, K87A and R90A, and others reported in Fig. 2f and Fig. 3e, this is again mentioned for clarity in the footnotes of each Table.

- d. Table 5 for Extended Fig. 9a, the WT-Rv3806c PRTase assays were performed alongside the mutants R240C, W175C, F176L, A237V, K174T, A249G.

Also, we have updated the main manuscript referring to the Extended Date XLS File Tables 1 to 5, and accordingly the Figure Legends:Fig. 1c : (c) [¹⁴C] radiolabeled cell-free PRTase activity of purified Rv3806c protein. Data presented are mean +SD calculated from three independent experiments (see Extended Data XLS File Table 1).

Fig. 2f: PRTase activities of WT-Rv3806c and L231W mutant. The data presented for the WT-Rv3806c and L231W mutant lane is mean +SD calculated from three independent experiments (see Extended Data XLS File Table 2)

Fig. 3e: PRTase analysis of polar interaction sites of DP shown in 3d. Data presented for all mutants are mean +SD calculated from three independent experiments (see Extended Data XLS File Table 3).

Fig. 4d: PRTase activity of WT-Rv3806c and mutant Rv3806c, mutated residues were grouped by their interaction sites on PRPP-Mg²⁺. Data presented for all mutants are mean +SD calculated from three independent experiments (Extended Data XLS File Table 4).

Extended Data File Fig. 1c: Radiometric-TLC analysis of PRTase activity using radiolabeled P[¹⁴C]RPP. The DPP[¹⁴C]R product catalyzed by Rv3806c is shown resolved from [¹⁴C]ribose phosphate at the origin which was generated in situ through chromatography by TLC as shown (see Extended Data XLS File Table 1).

Extended Data File Fig. 9a. PRTase enzymatic activity analysis of clinical ethambutol resistant Rv3806c mutations. Data presented are mean +SD calculated from three independent experiments for the mutant Rv3806c proteins, and for the WT-Rv3806c proteins from two independent experiments (see Extended Data XLS File Table 5).

Decision Letter, second revision:

Message Our ref: NMICROBIOL-23051132B

:
15th January 2024

Dear Dr. Rao,

Thank you for your patience as we've prepared the guidelines for final submission of your Nature Microbiology manuscript, "Structure and mechanism of a membrane-bound

1phosphoribosyltransferase central to *M. tuberculosis* cell wall biosynthesis" (NMICROBIOL-23051132B). Please carefully follow the step-by-step instructions provided in the attached file, and add a response in each row of the table to indicate the changes that you have made. Please also check and comment on any additional marked-up edits we have proposed within the text. Ensuring that each point is addressed will help to ensure that your revised manuscript can be swiftly handed over to our production team.

In recognition of the time and expertise our reviewers provide to Nature Microbiology's editorial process, we would like to formally acknowledge their contribution to the external peer review of your manuscript entitled "Structure and mechanism of a membrane-bound phosphoribosyltransferase central to *M. tuberculosis* cell wall biosynthesis". For those reviewers who give their assent, we will be publishing their names alongside the published article.

Nature Microbiology offers a Transparent Peer Review option for new original research manuscripts submitted after December 1st, 2019. As part of this initiative, we encourage our authors to support increased transparency into the peer review process by agreeing to have the reviewer comments, author rebuttal letters, and editorial decision letters published as a Supplementary item. When you submit your final files please clearly state in your cover letter whether or not you would like to participate in this initiative. Please note that failure to state your preference will result in delays in accepting your manuscript for publication.

Cover suggestions

COVER ARTWORK: We welcome submissions of artwork for consideration for our cover. For more information, please see our guide for cover artwork.

Nature Microbiology has now transitioned to a unified Rights Collection system which will allow our Author Services team to quickly and easily collect the rights and permissions required to publish your work. Approximately 10 days after your paper is formally accepted, you will receive an email in providing you with a link to complete the grant of rights. If your paper is eligible for Open Access, our Author Services team will also be in touch regarding any additional information that may be required to arrange payment for your article.

Please note that *Nature Microbiology* is a Transformative Journal (TJ). Authors may publish their research with us through the traditional subscription access route or make their paper immediately open access through payment of an article-processing charge (APC). Authors will not be required to make a final decision about access to their article until it has been accepted. Find out more about Transformative Journals

Best regards,

Reviewer #2:

Remarks to the Author:

All concerns are addressed. Congratulations to the very nice work. I just noticed one grammatical issue at Line 135: "consistence" should read "consistent".

Reviewer #4:

None

Final Decision Letter:

Message: 7th February 2024

Dear Professor Rao,

I am pleased to accept your Article "Structural analysis of phosphoribosyltransferase-mediated cell wall precursor synthesis in *Mycobacterium tuberculosis*" for publication in *Nature Microbiology*. Thank you for having chosen to submit your work to us and many congratulations.

Over the next few weeks, your paper will be copyedited to ensure that it conforms to *Nature Microbiology* style. We look particularly carefully at the titles of all papers to ensure that they are relatively brief and understandable.

Please note that *Nature Microbiology* is a Transformative Journal (TJ). Authors may publish their research with us through the traditional subscription access route or make their paper

4immediately open access through payment of an article-processing charge (APC). Authors will not be required to make a final decision about access to their article until it has been accepted. Find out more about Transformative Journals

With kind regards,